# ER-residential Nogo-B accelerates NAFLD-associated HCC mediated by metabolic reprogramming of oxLDL lipophagy

Yuan Tian[1,12], Bin Yang[1,2,12], Weinan Qiu[1,2], Yajing Hao[2,3], Zhenxing Zhang[1,2], Bo Yang[1,2], Nan Li[4], Shuqun Cheng[4], Zhangjun Lin[1], Yao-cheng Rui[5], Otto K.W. Cheung[6], Weiqin Yang[6], William K.K. Wu[7,8], Yue-Sun Cheung[9], Paul B.S. Lai[9], Jianjun Luo[3], Joseph J.Y. Sung[7,10], Runsheng Chen[3], Hong-Yang Wang[4,11], Alfred S.L. Cheng[6,7] & Pengyuan Yang[1,2,11]

Non-alcoholic fatty liver disease (NAFLD) is the hepatic manifestation of the metabolic syndrome that elevates the risk of hepatocellular carcinoma (HCC). Although alteration of lipid metabolism has been increasingly recognized as a hallmark of cancer cells, the deregulated metabolic modulation of HCC cells in the NAFLD progression remains obscure. Here, we discovers an endoplasmic reticulum-residential protein, Nogo-B, as a highly expressed metabolic modulator in both murine and human NAFLD-associated HCCs, which accelerates high-fat, high-carbohydrate diet-induced metabolic dysfunction and tumorigenicity. Mechanistically, CD36-mediated oxLDL uptake triggers CEBPβ expression to directly upregulate Nogo-B, which interacts with ATG5 to promote lipophagy leading to lysophosphatidic acid-enhanced YAP oncogenic activity. This CD36-Nogo-B-YAP pathway consequently reprograms oxLDL metabolism and induces carcinogenetic signaling for NAFLD-associated HCCs. Targeting the Nogo-B pathway may represent a therapeutic strategy for HCC arising from the metabolic syndrome.

[1] Key Laboratory of Infection and Immunity of CAS, CAS Center for Excellence in Biomacromolecules, Institute of Biophysics, Chinese Academy of Sciences, 100101 Beijing, China. [2] University of Chinese Academy of Sciences, 100049 Beijing, China. [3] Key Laboratory of RNA Biology of CAS, Institute of Biophysics, Chinese Academy of Sciences, 100101 Beijing, China. [4] Eastern Hepatobiliary Surgery Hospital, Second Military Medical University, 200433 Shanghai, China. [5] Department of Pharmacology and School of Pharmacy, Second Military Medical University, 200433 Shanghai, China. [6] School of Biomedical Sciences, The Chinese University of Hong Kong, 999077 Hong Kong, China. [7] State Key Laboratory of Digestive Disease, The Chinese University of Hong Kong, 999077 Hong Kong, China. [8] Department of Anaesthesia and Intensive Care, The Chinese University of Hong Kong, 999077 Hong Kong, China. [9] Department of Surgery, The Chinese University of Hong Kong, 999077 Hong Kong, China. [10] Department of Medicine and Therapeutics, The Chinese University of Hong Kong, 999077 Hong Kong SAR, China. [11] National Center for Liver Cancer, Second Military Medical University, 200433 Shanghai, China. [12] These authors contributed equally: Yuan Tian, Bin Yang. Correspondence and requests for materials should be addressed to A.S.L.C. (email: alfredcheng@cuhk.edu.hk) or to P.Y. (email: pyyang@ibp.ac.cn)

Hepatocellular carcinoma (HCC) is the fifth most prevalent primary cancer and the third leading cause of cancer-related mortality worldwide. HCC is driven by etiological factors, including hepatitis B or C virus, carcinogen/toxin exposure, and increasingly, metabolic disorders. Despite a drop in new HBV and HCV infections, the incidence of HCC in patients with metabolic disorders, clinically termed non-alcoholic fatty liver disease (NAFLD), has been rising[1]. NAFLD is characterized by hepatic fat accumulation and is closely associated with central obesity, diabetes, and other features of the metabolic syndrome[2]. NAFLD encompasses a wide disease spectrum ranging from simple steatosis to non-alcoholic steatohepatitis (NASH), which may in turn develop into cirrhosis, end-stage liver disease, or HCC[3]. Although NAFLD has become a serious global health threat, the molecular mechanisms underlying NAFLD-associated HCC remain obscure.

The endoplasmic reticulum (ER) stress response has recently been reported to play a crucial role in the development of NAFLD and to be linked to dysfunction of lipolysis, insulin resistance, inflammation, and cell apoptosis[4]. Clinical histological assessments describe NAFLD as a macrovesicular steatosis in which the nucleus is displaced by large lipid droplets (LD) to the edge of the cell[5]. The ER regulates lipid metabolic processes. Imbalanced intracellular lipid homeostasis and changes in lipid/cholesterol content consequentially disturb the ER structure and cause ER stress[6]. Lipolysis has been classically recognized to occur through the actions of the cytosolic neutral lipase that hydrolyzes LD-stored triglycerides (TGs) into free fatty acids (FFAs) and glycerol. Recognition of similarities in regulation and function of lipolysis and macroautophagy led to the identification of a specific form of autophagy termed lipophagy that degrades LD-sequestered lipids[7]. Considering the important role of lipid metabolism in liver homeostasis, dysfunctional lipophagy may be a contributing factor in hepatic lipid metabolism during the development of NASH and HCC[8]. On the other hand, the breakdown of stored lipids from LDs into FFA not only provides energy to the cell, but also modulates other cellular processes, such as activating carcinogenic signaling pathways that contribute to cancer development[8]. Growing evidence supports the notion that macroautophagy can be used by hepatic cancer cells for tumor progression. In fact, increased levels of macroautophagy markers, such as light chain 3 (LC3) in hepatocarcinoma, have been associated with poor prognoses and higher rates of recurrence after surgery[9,10]. However, the role of LD metabolism, especially lipophagy, in HCC progression is unclear. The molecular mechanisms linking the disordered metabolic programming to liver tumorigenesis remain elusive.

In the present study, we profiles and identifies an ER-associated Reticulons (RTN) family gene, Nogo-B, which is highly expressed in both murine and human NAFLD-associated HCC tissues. We discovers the hitherto unrecognized function of Nogo-B as a liver metabolic regulator that reprograms the lipophagy-mediated LD degradation to enhance oncogenic Yes-associated protein (YAP) activity. These findings demonstrate the functional significance of Nogo-B and provide a strong impetus for therapeutic interventions.

## Results

**Nogo-B is highly expressed and promotes tumorigenesis in HCC.** Regarding ER stress being commonly stimulated during NAFLD and HCC progression and the dynamic crosstalk between ER and LD[11], we focused on the key regulators of lipid homeostasis that are anchored to the ER. To characterize the ER-located proteins involved in HCCs accompanied by signature metabolic disorders, gene expression profiling covering the 103

ER network genes (Supplementary Table 1) was performed in three pairs of tumors and adjacent normal tissues from a murine NASH-associated HCC model induced by hepatocarcinogen and high-fat high-carbohydrate (HFHC) diet, in parallel with human HCC subjects in TCGA database (Fig. 1a). Ten genes were differentially expressed (2 fold-difference) in tumors and paired adjacent normal tissues (Fig. 1b and Supplementary Fig. 1A). Among the top upregulated genes, Nogo-B attracted our attention because it was previously reported to regulate liver cancer proliferation through the IL-6/STAT3 pathway[12]. However, little is known about its ER-related function in tumorigenesis and related metabolic processing. To confirm the upregulation of Nogo-B, we examined Nogo-B mRNA and protein levels using 11 additional paired tumors and adjacent noncancerous livers from murine NASH-associated HCC by quantitative reverse transcription PCR (qRT-PCR) (Supplementary Fig. 1B) and western blot analysis (Fig. 1c). Our data demonstrate that Nogo-B is strongly upregulated in murine NASH-associated HCC tissues.

To further demonstrate the significance of Nogo-B in liver cancer, we retrospectively analyzed 103 network gene transcripts in the TCGA database of 360 HCC samples and 50 adjacent non-tumor liver tissues from HCC patients (Fig. 1d and Supplementary Fig. 1C, D). Consistently, Nogo-B was one of the outstanding genes according to the following criteria: Firstly, hepatic gene expression was high enough for further study (Count per million, CPM å 1); Secondly, the change in tumors compared with adjacent normal tissues was >2-fold (Log2 (FC) å 1 or < −1); Thirdly, the expression difference between tumors and normal tissues was statistically significant ($p < 0.01$). Additionally, we checked the Nogo-B gene amplification level in HCC patients using the cBioportal and Oncomine websites and found that the DNA copy number of Nogo-B in HCC tumors was higher compared with adjacent non-tumor liver tissues or normal livers from healthy donors in three cohorts. In addition, the DNA copy number of Nogo-B was positively correlated with mRNA expression (Supplementary Fig. 1E, G), thus confirming its clinical significance in HCC oncogenesis.

To assess the functional role of Nogo-B in oncogenesis, colony formation and cell proliferation assays were performed. Ectopic expression of Nogo-B in two cell lines with low Nogo-B expression, the immortalized hepatocyte LO2 and HCC cell line SMMC-7721 (Fig. 1e and Supplementary Fig. 1H), promoted cell growth (Fig. 1f and Supplementary Fig. 1I), while lentiviral-mediated knockdown of Nogo-B in SK-Hep1 and MHCC97H cells resulted in remarkable growth inhibition (Fig. 1g and Supplementary Fig. 1J). To investigate the effect of Nogo-B on tumorigenesis of HCC cells in vivo, we employed Nogo-B stably transfected cells (Nogo-B) or empty vector-transfected cells (vector) and injected the cells into the dorsal flanks of nude mice ($n = 5$ per group). After subcutaneous injection, LO2 cells with Nogo-B overexpression displayed faster and more sustainable tumor growth in the xenograft model compared with empty vector control group (Fig. 1h and Supplementary Fig. 1K). Conversely, MHCC97H cells that stably expressed short-hairpin RNA (shRNA) targeting Nogo-B (shNogo-B) showed significantly reduced tumor growth compared with non-targeting shRNA-transfected cells (shCtrl) (Fig. 1i and Supplementary Fig. 1L). We further verified these in vivo findings using another orthotopic model in which 1.0 mm$^3$ subcutaneous tumor tissues were orthotopically implanted into mouse livers. Both the tumor volume and weight of the Nogo-B orthografts were increased while Nogo-B knockdown orthografts were decreased when compared with vector or control orthografts, respectively (Fig. 1j–k). Taken together, these in vitro and in vivo results suggest that Nogo-B exhibits strong oncogenic properties in HCC.

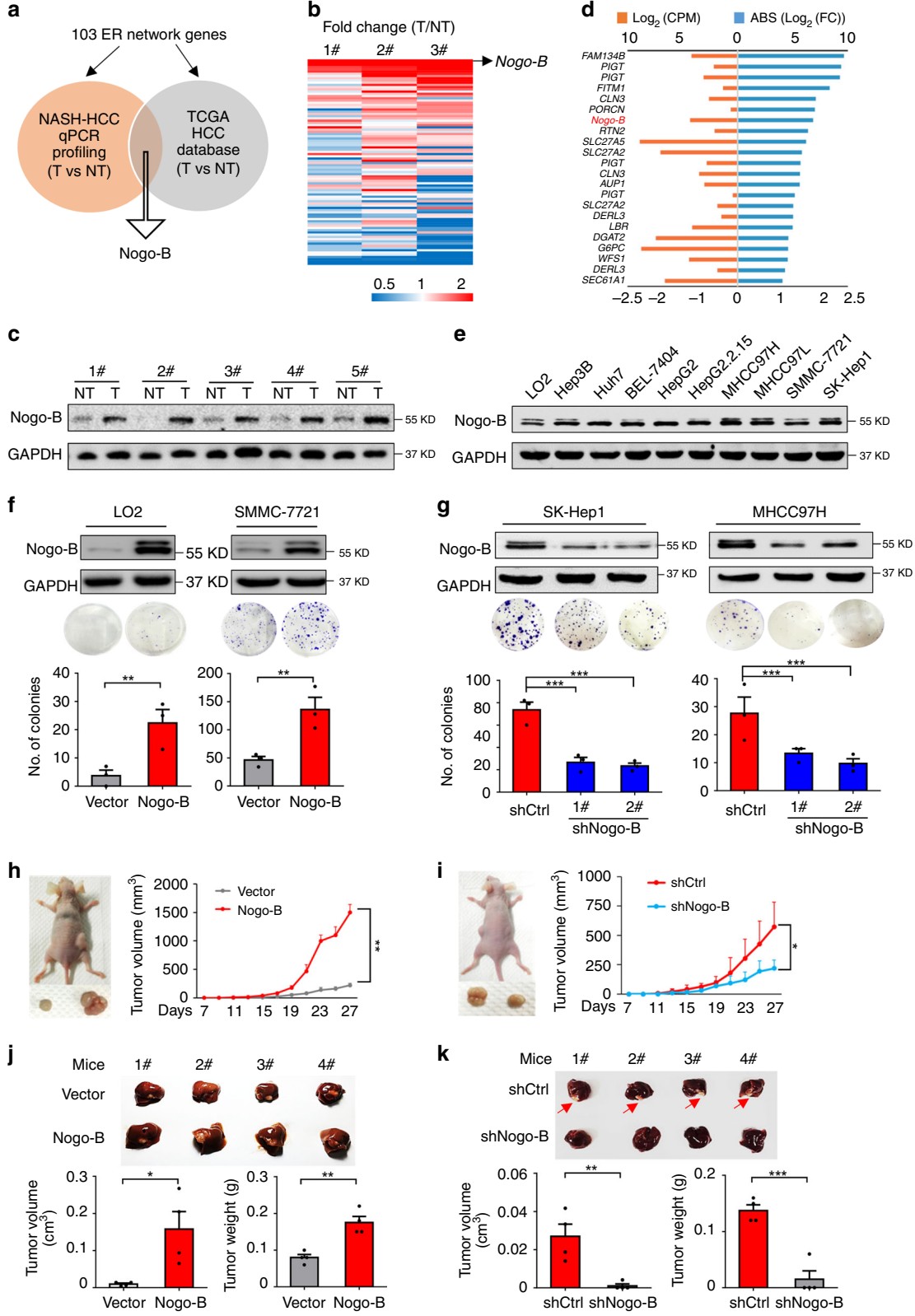

**Nogo-B is controlled by oxLDL-CD36-CEBPβ cascade.** To explore the underlying molecular mechanisms of Nogo-B upregulation, we compared the Nogo-B expression level in tumors from HCC patients with different etiological factors. Interestingly, the expression of Nogo-B was more remarkably enhanced in NAFLD-associated HCCs than that in HBV-associated HCCs (Fig. 2a), suggesting a possible role of Nogo-B in metabolic

dysregulation. To further elucidate the metabolic modulators in Nogo-B expression, a PCR Array that included 85 fatty liver-related genes (Supplementary Table 2) was performed in para-tumor and tumor tissues from a murine NASH-associated HCC model. The cluster of differentiation 36 (*Cd36*) was identified as the most stimulated gene (Fig. 2b), which was also confirmed using additional normal and tumor tissues by qRT-PCR and

**Fig. 1** Nogo-B is highly expressed and promotes tumorigenesis in HCC. **a** ER network genes profiling strategy. **b** Heatmap representation of the fold change of ER-residential genes expression in three paired tumors (T) and noncancerous livers (NT) from murine NASH-promoted HCC mice. **c** Western blot analysis of Nogo-B expression in paired tumors (T) and adjacent noncancerous livers (NT) from HFHC-treated mice. **d** Fold change and basal expression level of ER-residential genes in the TCGA database. **e** Western blot analysis of Nogo-B expression in normal liver cell line LO2 and multiple HCC cell lines. **f, g** Colony formation assay of (**f**) LO2 and SMMC-7721 cells transfected with empty control (vector) and Nogo-B expression plasmid (Nogo-B), and **g** SK-Hep1 and MHCC97H cells infected with lenti-shCtrl (shCtrl) or lenti-shNogo-B (shNogo-B) for 14 days. **h, i** Representative tumor images (left) and tumor volumes (right) of xenografts derived from **h** LO2 cells stably transfected with empty control (vector) and Nogo-B expression plasmid (Nogo-B), and **i** MHCC97H cells stably transfected with control (shCtrl) or Nogo-B shRNA (shNogo-B) ($n = 5$ in each group). In all, $5 \times 10^6$ cells were subcutaneously injected into the right and left flanks of the mice. The tumors were collected after 4 weeks. **j, k** Liver tumor images (top), tumor volume and tumor weight (bottom) of orthotopic models derived from **j** control and Nogo-B overexpressing LO2 cells, and **k** control and Nogo-B-knockdown MHCC97H cells ($n = 4$ in each group). In all, 1.0 mm³ tumor pieces were implanted into the left liver lobe of each mouse. The mice were sacrificed after 5 weeks. (Data are presented as mean ± SEM of three independent experiments in **f** and **g**. *$p < 0.05$, **$p < 0.01$, ***$p < 0.001$. Source data are provided as a Source Data file)

western blot analysis (Fig. 2c, d). A well-known scavenger receptor, CD36 has been reported as a potential diagnostic and treatment target in NAFLD progression for hepatic inflammation via mediating uptake of FFA and some modified low-density lipoprotein, especially for oxidized low-density lipoprotein (oxLDL)[13,14]. To test the role of CD36-mediated lipid uptake in NAFLD-associated carcinogenesis, we determined the LDL, oxLDL and FFA levels in NASH-associated HCC mice. The plasma oxLDL and FFA concentrations were significantly increased in HFHC-fed mice than in low-fat diet (LFD)-fed mice (Fig. 2e and Supplementary Fig. 2A, B). Notably, oxLDL-treated cells exhibited CD36 cell membrane translocation and co-localization with oxLDL (Supplementary Fig. 2C), leading to upregulation of Nogo-B expression, an effect, which was not observed in oleic acid treatment (Fig. 2f and Supplementary Fig. 2D–F). Additionally, the expression of other oxLDL receptors, including Sra, Sr-b1, Srec and Lox1, was also determined in murine NAFLD-associated HCCs, and there was no consistent alteration in HFHC mice and tumor tissues (Supplementary Fig. 2G). To explore the molecular mechanism, we analyzed the transcription factor ChIP-seq database and identified CCAAT/enhancer-binding protein β (CEBPβ), a downstream target of oxLDL reported in atherosclerosis, a chronic inflammatory disease and lipid metabolism disorder[15], as a potential regulator of Nogo-B (Supplementary Fig. 2H). We employed the short-interfering RNA (siRNA) approach to knockdown CEBPβ in MHCC97H cells and found a reduced *Nogo-B* promoter occupancy by CEBPβ (Fig. 2g). Nogo-B transcript and protein levels were also downregulated by siRNAs of CEBPβ (Fig. 2h, i). Moreover, knockdown of CEBPβ blocked the oxLDL-induced Nogo-B expression in SMMC-7721 cells (Fig. 2j). Concordant with Nogo-B upregulation, Cd36 and Cebpβ expressions were simultaneously increased and positively correlated with Nogo-B expression in murine HFHC-promoted HCC development (Fig. 2k and Supplementary Fig. 2I). Collectively, these data indicate that Nogo-B is upregulated upon oxLDL-stimulated CEBPβ activation in NAFLD-associated HCC.

**Nogo-B enhances NASH-associated carcinogenesis.** We then investigated the functional role of Nogo-B in NASH-associated carcinogenesis. In a dietary NASH-associated HCC model, we performed lentiviral-mediated knockdown of Nogo-B in the liver (Fig. 3a). Notably, knockdown of Nogo-B in the livers of obese mice (Fig. 3b, c) significantly reduced over 60% of NASH-associated tumor multiplicity and weight (Fig. 3d–f and Supplementary Fig. 3A). The reduced tumorigenicity was associated with a significant reduction in hepatic steatosis, hepatocyte ballooning, hepatocellular lipid accumulation, and hepatic inflammation (Fig. 3d–f and Supplementary Fig. 3B). Consistently, glucose sensitivity measured by the intraperitoneal glucose tolerance test (IPGTT) was improved by Nogo-B downregulation

(Fig. 3g). The lipids concentrations, including TG, FFA, and Cholesterol were increased in the HFHC dietary NAFLD-associated HCC model. Notably, knockdown of Nogo-B in the livers of obese mice significantly reduced the levels of these lipids except oxLDL (Fig. 3h–i and Supplementary Fig. 3C). These findings indicate that Nogo-B enhances carcinogenesis, possibly through metabolic modulations.

**Nogo-B promotes lipid droplet degradation in HCC cells.** As hepatic lipid overload and accumulation are major characteristics of NAFLD[16], we investigated the role of Nogo-B on lipid accumulation. Unexpectedly, we found there was much less lipid accumulation in the tumorous areas compared with that in the para-tumorous areas from obese mice, which showed a negative correlation with the Nogo-B level (Fig. 4a and Supplementary Fig. 4A). Considering the high energy and nutrient demands of cancer cells, we therefore speculated that Nogo-B might promote hepatocarcinogenesis by consuming excessive lipids as nutrient supplies and inducing metabolic reprogramming. To verify that Nogo-B resides on LDs in the hepatocyte, we first isolated LDs from mice that fasted overnight using subcellular fractionation and found Nogo-B was expressed in LDs in primary hepatocytes (Supplementary Fig. 4B, C). We then confirmed the co-localization of Nogo-B with LDs by determining fluorescently-tagged Nogo-B in LO2 cells (Fig. 4b and Supplementary Fig. 4D). Furthermore, LO2 cells expressing Nogo-B exhibited a stronger capacity to mobilize stored lipids when subjected to low-serum starvation for 24 h post-oxLDL loading (Fig. 4c). Conversely, MHCC97H cells stably transfected with shNogo-B showed impaired LD utilization (Fig. 4d and Supplementary Fig. 4E). Consistent with the negative correlation between Nogo-B and lipid accumulation in vivo, Nogo-B knockdown induced greater expression of the lipid marker PLIN2 in vitro (Supplementary Fig. 4F, G). Interestingly, Nogo-B overexpression or depletion did not affect LD formation as demonstrated by the fact that no difference in the amount of LD was observed between Nogo-B and vector cells (Fig. 4e, f). LD breakdown leads to degradation of TGs into FFAs, which is a critical step during cellular metabolic reprograming and energy support[17]. The promoted LD degradation resulted in increased FFAs and decreased TGs in Nogo-B-overexpressing LO2 and SMMC-7721 cells (Fig. 4g and Supplementary Fig. 4H), while starvation-induced metabolite conversion was blocked by Nogo-B knockdown in SK-Hep1 and MHCC97H cells (Fig. 4h and Supplementary Fig. 4I). Because of the critical role of ER during LD formation, we investigated whether the ER retention of Nogo-B is required for its promoted effect on LD degradation. Ectopic expression of the ER-motif-deficient Nogo-B (Nogo-B-d38)[18] in LO2 cells (Fig. 4i and Supplementary Fig. 4J) abolished the lipid degradation effect of wild-type Nogo-B (Fig. 4j). It has been reported that a transmembrane protein, Nogo-B receptor (NgBR), recognizes the other terminal (RHD

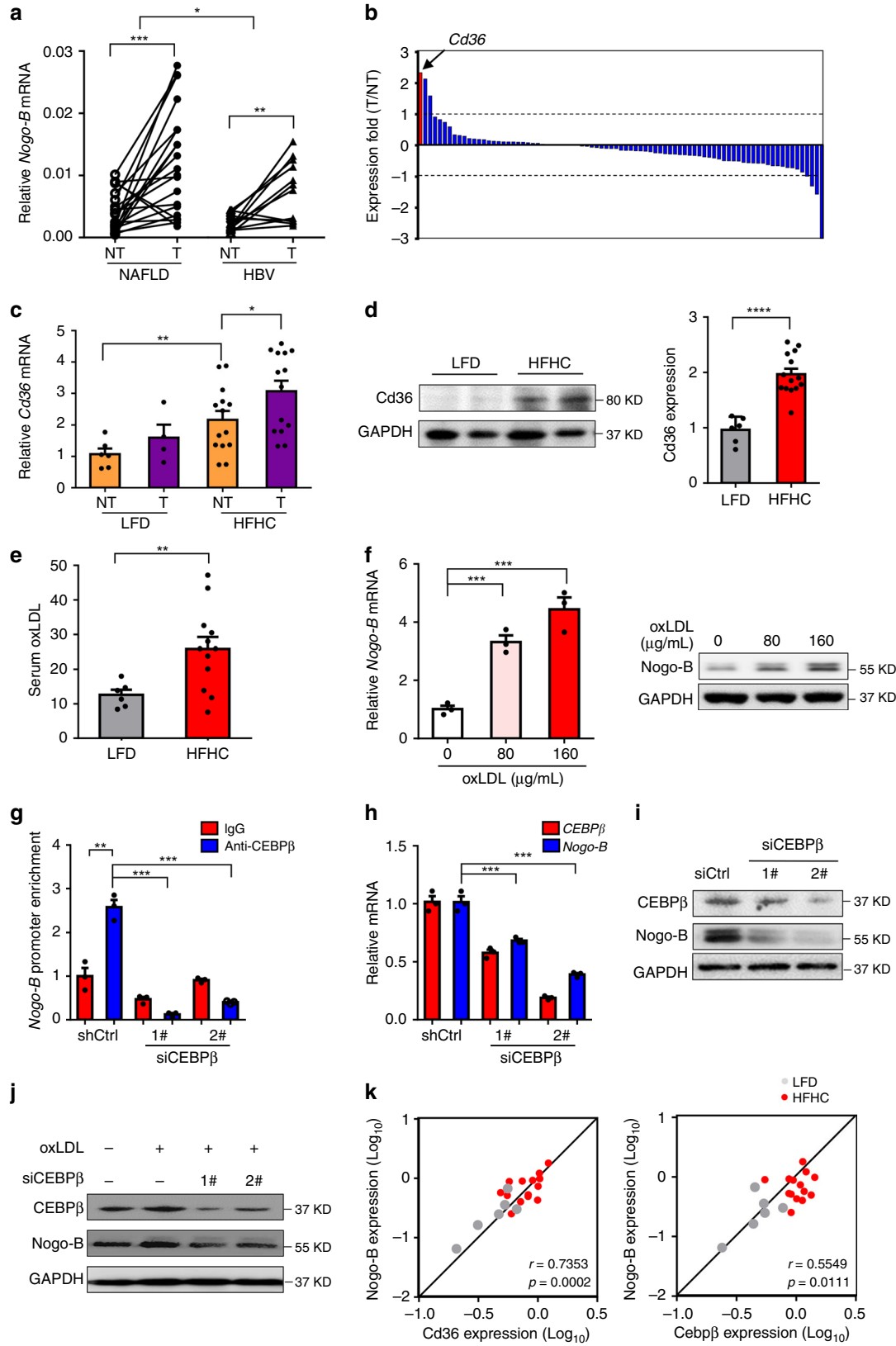

domain) of Nogo-B and could mediate Nogo-B-mediated chemotaxis and development of angiogenesis, and is also involved in hepatic lipogenesis[19,20]. Therefore, in order to test whether function of Nogo-B in tumorigenesis is mediated through an NgBR-mediated autocrine mechanism, we performed shRNA-mediated knockdown of NgBR and found shNgBR did not alter the colony formation of Nogo-B-expressing LO2 and SMMC-7721 cells (Supplementary Fig. 4K), indicating independence of NgBR. Additionally, both ectopic expression and knockdown experiments showed that Nogo-B did not affect the expressions of triglyceride lipase and hormone-sensitive lipase involved in TG hydrolysis[21] (Supplementary Fig. 4L). Taken together, these data

**Fig. 2** Nogo-B is enhanced by the oxLDL-CD36-CEBPβ cascade. **a** qRT-PCR analysis of *Nogo-B* mRNA expression in tumors (T) and adjacent normal tissues (NT) from NAFLD-associated ($n = 20$) and HBV-associated ($n = 12$) HCC patients, respectively. **b** PCR Array of fatty liver-related genes in two paired tumors (T) and adjacent normal tissues (NT) from murine NASH-associated HCC model. **c** qRT-PCR analysis of *Cd36* mRNA expression in livers from LFD- ($n = 6$) and HFHC-fed ($n = 14$) mice. **d** Representative western blot images (left) and quantification (right) of Cd36 expression in the livers of mice described in **c**. **e** Serum oxLDL level of mice described in livers from LFD- ($n = 6$) and HFHC-fed ($n = 12$) mice. **f** qRT-PCR (left) and western blot (right) analysis of Nogo-B expression in SMMC-7721 cells stimulated with oxLDL at indicated dosages for 24 h. **g–i** CEBPβ directly upregulates Nogo-B expression as shown by **g** ChIP-PCR, **h** qRT-PCR, and **i** western blot analysis. MHCC97H cells were transfected with CEBPβ siRNAs for 48 h. **j** Western blot analysis of Nogo-B expression in SMMC-7721 cells transfected with CEBPβ siRNAs followed by oxLDL treatment (80 μg/mL) for 24 h. **k** Correlation of protein levels of Cd36 and Nogo-B (left), and Cebpβ and Nogo-B (right) in the livers of mice described in **c**. (Data are presented as mean ± SEM of three independent experiments in **f–h**. \*$p < 0.05$, \*\*$p < 0.01$, \*\*\*$p < 0.001$. Source data are provided as a Source Data file)

demonstrate that the ER location of Nogo-B is required for its role in promoting LD breakdown to degrade TGs into FFAs in HCC cells.

**Nogo-B promotes lipophagy in HCC cells**. Since lipophagy represents a major mechanism for LD degradation and most cancer cells exhibit higher levels of basal autophagy than normal cells[7,22], we postulated that Nogo-B-promoted LD breakdown is lipophagy-dependent. We first detected lipophagy by co-localization of LC3 and LD in oxLDL-treated LO2 hepatocytes and RAW264.7 macrophages (Fig. 5a and Supplementary Fig. 5A and B). To better indicate the lipophagy progress, we performed electron microscopy to clarify that autophagosomes enveloping LDs, and found double-membrane vesicles analogous to autophagosomes (arrow heads) around LDs (arrows) and degradative structures enriched in LDs (asterisks, Supplementary Fig. 5C), which is consistent with previous report of lipophagy[23]. RAB7 has been implicated in the initiation of autophagy-centric LD catabolism by mediating docking of autophagosomes (AP) and lysosomes to LDs during lipophagy[24]. To verify the role of lipophagy in nutrient supply in HCC cells, we further investigated the location of RAB7 and LC3 on the LD surface. In oxLDL-loaded SK-Hep1 and MHCC97H cells, most of the LipidTOX-stained LDs contained RAB7 or LC3. In contrast, cells expressing shNogo-B exhibited almost no interactions between LD with RAB7 or LC3 (Fig. 5b and Supplementary Fig. 5D). As lipophagy is an autophagic process, we also analyzed whether modulation of Nogo-B expression affects AP functions in HCC cells. To test this, we transfected LO2 and SMMC-7721 cells with a dually fluorescent mCherry-EGFP-LC3 reporter, which can be used to monitor AP-to-autolysosome maturation by virtue of the pH sensitivity of EGFP. As shown in Fig. 5c and Supplementary Fig. 5E, F, the numbers of both the autophagosomes (mCherry +, EGFP + ) and autolysosomes (mCherry +, EGFP-) were increased by Nogo-B overexpression. Moreover as expected, Nogo-B overexpressing stable cell line showed obvious higher Nogo-B expression than vector control cell line (Supplementary Fig. 5G). This observation was further confirmed by Nogo-B-induced upregulation of LC3-II, ATG7, conjugated form of ATG5 (ATG5-ATG12) and monomeric form of ATG5, as well as the downregulation of p62 in LO2 and SMMC-7721 cells. Additionally, after treatment of these stable cells with autophagy inhibitors wortmannin (wort) and chloroquine (CQ), We found that wortmannin inhibited Nogo-B induced p62 degradation and LC3 conversion while there were a slightly higher LC3-II level and the same p62 level in Nogo-B overexpressing stable cells compared with control vector cells after CQ treatment (Fig. 5d and Supplementary Fig. 5H, I). These data strengthened that Nogo-B promoted autophagy flux in HCC cells. Moreover, oxLDL-induced Nogo-B expression also enhanced autophagy levels in a dose-dependent manner (Supplementary Fig. 5J).

We next determined the mechanism by which Nogo-B regulates the autophagic catabolism of LDs. We speculated that

as an ER-residential protein, Nogo-B might recruit the phagophore components for autophagy flux to the LD surface. Using immunofluorescence and co-immunoprecipitation, we found a physical interaction between Nogo-B and ATG5, which are both localized in ER and cytoplasm, in Nogo-B expressing LO2 cells, MHCC97H cells and BEL-7404 cells (Fig. 5e–g and Supplementary Fig. 5K–L). Moreover, the interaction was disrupted by overexpression of ER-motif-deficient Nogo-B (Fig. 5h). Additionally, we found higher Atg5-Atg12, Atg7 and LC3-II but lower p62 levels in the HFHC diet as compared to the control diet group (Supplementary Fig. 5M), and the changes of autophagy markers were remarkably reversed by Nogo-B downregulation. To further clarify the autophagy level in NAFLD-associated HCC, we performed western blot using the tumorous and adjacent non-tumorous tissues of the mouse model. Consistently, HCC tumors exhibited higher Atg5-Atg12, Atg7 and LC3-II, and lower p62 levels than non-tumors (Supplementary Fig. 5N). These results consolidate our observation that Nogo-B promotes autophagy in NAFLD-HCC development. To further investigate the effect of lipophagy progress on Nogo-B-induced LD breakdown and cell growth, we performed LD staining using LipidTOX and colony formation assays on LO2 and SMMC-7721 cells co-transfected with vectors expressing Nogo-B and shATG5. Notably, Nogo-B ectopic expression significantly enhanced the breakdown of LDs and colony formation, which could be abrogated by ATG5 knockdown (Fig. 5I, j and Supplementary Fig. 5O–Q). These results suggest that Nogo-B promotes hepatocellular proliferation through lipophagy-mediated LD breakdown.

**Nogo-B stimulates YAP activity to induce carcinogenesis**. To elucidate the oncogenic mechanism, we employed a luciferase reporter-based cancer pathways array in LO2 cells to interrogate the signaling pathways that could be regulated by Nogo-B overexpression. Among the 11 pathways, Hippo signaling was most highly activated upon ectopic Nogo-B expression (Fig. 6a). This was further validated by the modulation of Nogo-B expression in other HCC cells (Supplementary Fig. 6A). Based on the functional role of Nogo-B in lipophagy of oxLDL, we further explored HPLC-MS to analyze the lipidomics in the livers from NASH-HCC mice. Importantly, among 16 significantly altered metabolites ($p < 0.05$) (Fig. 6b and Supplementary Table 3), lysophosphatidic acid (LPA), the most significantly reduced metabolite after Nogo-B knockdown, has been showed to regulate the Hippo pathway[25]. Since LPA could be catalyzed from lysophosphatidylcholine (LPC), a major activated component of oxLDL, we postulated that Nogo-B stimulated YAP activity through enhanced LPA levels. Concordant with the luciferase activity data, Nogo-B decreased the protein level of inactive (phosphorylated) YAP and increased the transcript levels of its downstream targets connective tissue growth factor (*CTGF*) and cysteine-rich protein 61 (*CYR61*) (Fig. 6c and Supplementary Fig. 6B). On the contrary, ablation of Nogo-B using shRNA in MHCC97H and SK-Hep1 cells reduced p-YAP levels and induced *CTGF* and *CYR61*

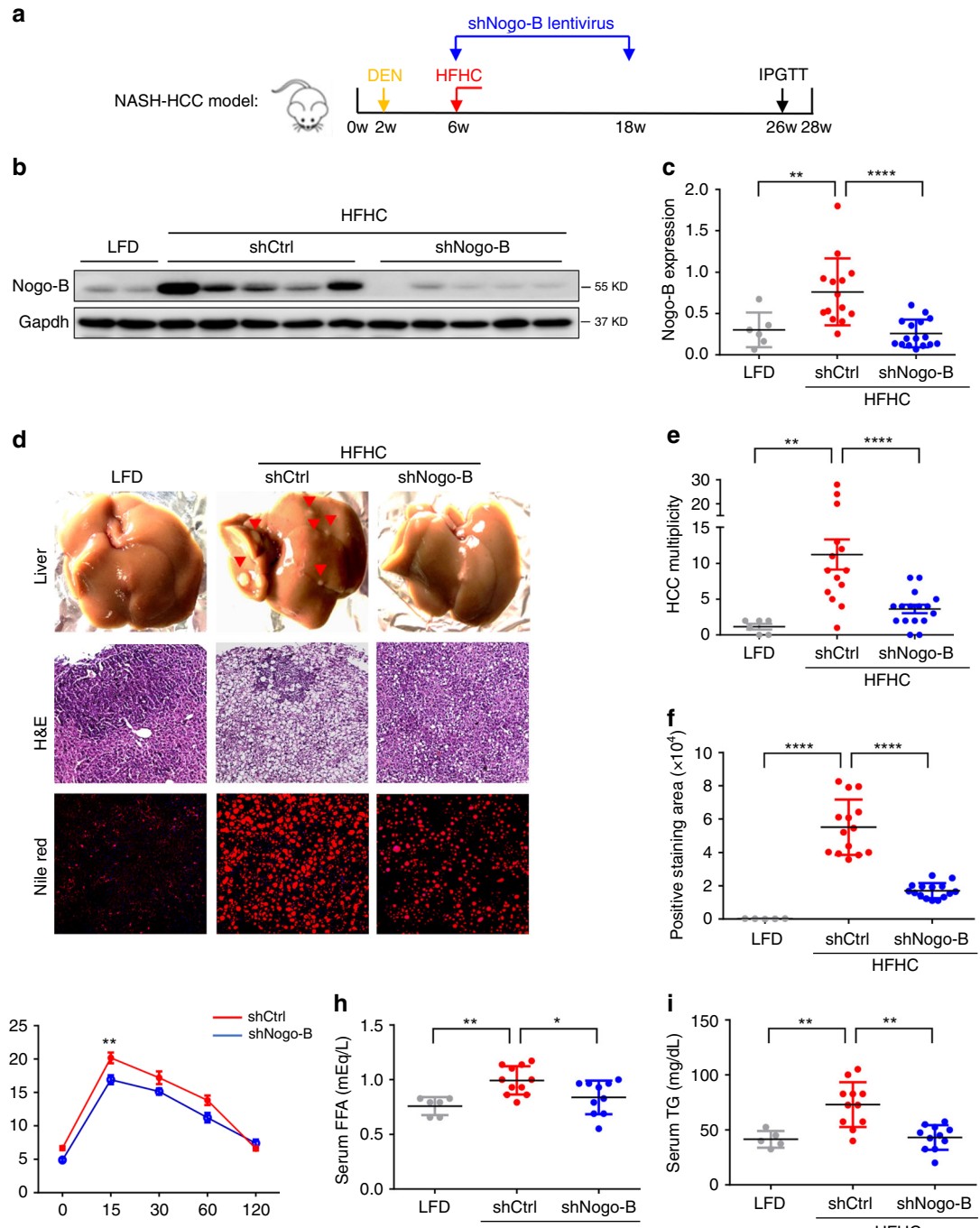

**Fig. 3** Knockdown of Nogo-B inhibits NASH and HCC progression. **a** Scheme of HFHC-promoted NASH-associated HCC model. **b**, **c** Representative (**b**) western blot images and (**c**) quantification of Nogo-B expression in the livers of DEN-treated and, LFD- or HFHC-fed mice administered with lentivirus expressing shCtrl or shNogo-B. Mice were sacrificed at 28 weeks of age. LFD/shCtrl: $n = 6$, HFHC/shCtrl: $n = 14$, HFHC/shNogo-B: $n = 16$.
**d** Representative liver pictures (top), H&E staining (middle) and Nile red staining (bottom) in the livers of LFD- and HFHC-fed mice with lentivirus administration. **e** The average tumor multiplicity of LFD- and HFHC-fed mice described in **c**. **f** The percentages of positively stained Nile red areas in the livers of LFD- and HFHC-fed mice described in **c**. **g** Blood glucose of 26-week-old HFHC-fed mice at indicated time-points after glucose injection in IPGTT analysis ($n = 12$ in each group). **h**, **i** Serum **h** FFA and **i** TG concentrations of LFD- and HFHC-fed mice described in **c**. (*$p < 0.05$, **$p < 0.01$, ****$p < 0.0001$. Source data are provided as a Source Data file)

expression (Fig. 6c and Supplementary Fig. 6B). Moreover, both intracellular and extracellular LPA and LPC concentrations were increased by Nogo-B ectopic expression in oxLDL-loaded LO2 and SMMC-7721 cells and decreased by Nogo-B knockdown in oxLDL-loaded MHCC97H and SK-Hep1 cells (Fig. 6d and Supplementary Fig. 6C, D). To determine whether YAP activity was

dependent on Nogo-B-induced LD degradation, the YAP phosphorylation level and downstream gene expression were detected in LO2 cells transfected with vectors expressing Nogo-B and siATG5. ATG5 knockdown interrupted Nogo-B induced dephosphorylation of YAP, as well as transcriptional levels of *CTGF* and *CYR61*, indicating that the role of Nogo-B for YAP activity was

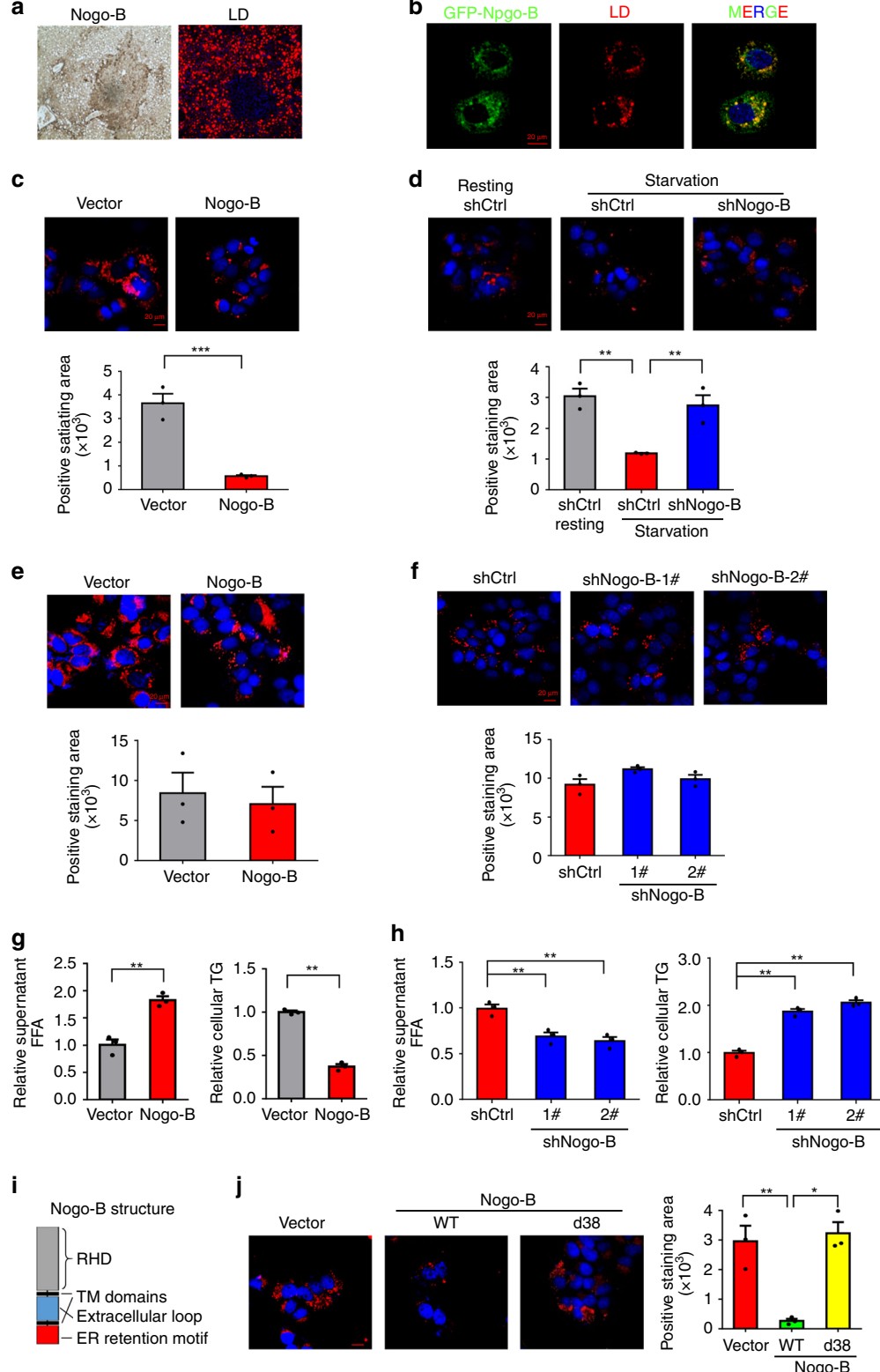

dependent on ATG5-involved lipophagy (Fig. 6e and Supplementary Fig. 6E). Functionally, the enhanced LPA level in Nogo-B-expressing LO2 cells was also disrupted upon ATG5 knockdown (Fig. 6f). To investigate whether the Nogo-B-enhanced LPA is specifically required for YAP activation, we pretreated hepatocytes with the inhibitor of autotaxin (ATX, PF-8380), a specific LPA-producing enzyme[26], to block the Nogo-B-mediated LPA production. ATX inhibitor abolished the Nogo-B-induced YAP

activation (Fig. 6g). Furthermore, the elevated YAP activity was confirmed in animal models, as lower expression of p-Yap and higher *Ctgf* and *Cyr61* levels were found in most tumor tissues compared with adjacent normal tumors in the NASH-associated HCC model (Fig. 6h and Supplementary Fig. 6F). Moreover, Nogo-B knockdown decreased the hepatic LPA concentrations, especially 22:6 LPA concentrations and activated Yap levels. In addition, Nogo-B expressions correlated strongly with LPA

**Fig. 4** Nogo-B promotes lipid droplet degradation in HCC cells. **a** Immunohistochemical staining of Nogo-B and Nile red staining of murine NASH-associated HCCs. **b** Co-localization of GFP-Nogo-B and LDs in LO2 cells transfected with Nogo-B for 48 h and incubated with oxLDL for 24 h followed by starvation. **c** Representative LipidTOX staining (top) and quantification of positively stained areas (bottom) of Nogo-B-overexpressing LO2 cells treated with oxLDL for 24 h followed by starvation in 0.1% FBS for 12 h. Cellular lipid droplets were stained using LipidTOX for 30 min. **d** Representative LipidTOX staining (top) and quantification of positively stained areas (bottom) of oxLDL-loaded MHCC97H cells stably transfected with shCtrl or shNogo-B with starvation or without starvation (resting) in 0.1% FBS for 12 h. **e, f** Representative LipidTOX staining (top) and quantification of positively stained areas (bottom) of **e** Nogo-B-overexpressing LO2 cells and **f** MHCC97H cells stably transfected with shCtrl or shNogo-B and treated with oxLDL for 24 h. **g, h** FFA concentration in supernatant (left) and TG concentration in cell lysate (right) extracted from **g** Nogo-B-overexpressing LO2 cells and **h** MHCC97H cells stably transfected with shCtrl or shNogo-B and treated with oxLDL for 24 h followed by starvation. **i** The domain organization and dissection of the human Nogo-B protein. **j** Representative LipidTOX staining (left) and quantification of positively stained areas (right) of wile-type Nogo-B (WT) or ER-motif-deficient Nogo-B (d38) ectopically expressed LO2 cells treated with oxLDL for 24 h followed by starvation in 0.1% FBS for 12 h. (Data are presented as mean ± SEM of three independent experiments in **c–h** and **j**. *$p < 0.05$, **$p < 0.01$, ***$p < 0.001$. Source data are provided as a Source Data file)

concentrations in the NAFLD-HCC model (Fig. 6i–k and Supplementary Fig. 6G–I). Collectively, these findings demonstrate that Nogo-B promotes tumorigenesis, at least partially, by LPA-mediated activation of YAP signaling.

**Enhanced Nogo-B cascade in clinical NAFLD-associated HCCs.**
To investigate the clinical relevance of our findings, we examined the protein levels of Nogo-B, CD36, CEBPβ, and p-YAP by western blot analysis in 16 pairs of human NAFLD-associated HCCs without either viral or alcoholic hepatitis, and 12 pairs of HBV-positive HCC patients. Compared with the paired adjacent normal liver tissues, concordant upregulation of Nogo-B, oxLDL, CD36, and CEBPβ and downregulation of p-YAP were detected in most NAFLD-associated HCC tissues, while there was no consistent expression pattern in HBV-associated HCC tissues, indicating that Nogo-B signaling was more activated in HCCs with metabolic disorders (Fig. 7a and Supplementary Fig. 7). Quantitative analysis further showed significant increases of Nogo-B, oxLDL, CD36, and CEBPβ and decrease of p-YAP in tumors from NAFLD patients (Fig. 7b). Importantly, correlation analysis further displayed significant positive associations between oxLDL and Nogo-B, oxLDL and CD36, CD36 and Nogo-B, as well as between CEBPβ and Nogo-B at protein level (Fig. 7c), demonstrating a strong activation of the oxLDL-CD36-CEBPβ cascade for Nogo-B induction in clinical NAFLD-associated HCC. To extend our findings to HCC patient prognosis, we analyzed *CD36*, *Nogo-B*, and *CYR61* expression in the TCGA dataset. Although the survival period data did not show statistically significant differences between low and high expression of the individual genes, we found that concomitant upregulation of the three genes was strongly associated with poor patient survival (Fig. 7d). Taken together, our findings in clinical specimens consolidate our findings that Nogo-B as a liver metabolic regulator reprograms the oxLDL lipophagy to enhance oncogenic YAP activity in NAFLD-associated HCCs (Fig. 7e).

**Discussion**
Accumulating reports have shown that obesity-related steatosis exerts a huge public health burden worldwide. As a central hub for lipid metabolism, liver diseases are closely related to metabolic disorders[27]. LD is the major organelle that regulates lipid homeostasis, and disturbance of LD-associated proteins induces metabolic diseases such as NAFLD[28]. Although there are several models for cytosolic LD biogenesis in cells, one shared feature is that LDs are formed in association with the ER. A number of membrane-bound proteins, mostly identified in animals and yeast, have been shown to be involved in the biogenesis of LDs and localized at the ER, mainly at ER-LD junctions[29,30]. In the present study, we have identified Nogo-B as a pivotal ER-resident oncogene in the development of NAFLD-associated HCC

through its role in promoting lipophagy-mediated LD turnover for oxLDL metabolism and subsequent LPA-stimulated YAP oncogenic activity (Fig. 7e).

Nogo-B is primarily known to play important roles in pathological vascular conditions in response to vascular injuries such as ischemia and atherosclerosis[31,32]. Nogo-B was reported to be highly expressed in hepatic stellate cells and to enhance liver fibrosis through facilitating the TGF-β signaling pathway[33]. Upregulation of Nogo-B was also found in HCCs to stimulate the IL-6/STAT3 pathway and alcoholic liver disease through the regulation of M1 polarization of Kupffer cells[12,34]. However, the molecular actions of Nogo-B in liver metabolism that induces tumorigenesis remain elusive. In this study, we demonstrated that Nogo-B is significantly upregulated in HCC, especially in NAFLD-associated HCC in both human clinical specimens and murine models. Our functional study using knockdown and ectopic expression approaches has confirmed the oncogenic role of Nogo-B in vitro and in vivo, and that Nogo-B promotes cancer cell growth through dysregulated metabolic programming. The dramatic inhibition of tumor numbers and weight by lentivirus-mediated Nogo-B downregulation in a HFHC-induced HCC model further establishes the strong oncogenic activity of Nogo-B during NAFLD progression. Consistent with a previous study that Nogo-B knockdown results in decreased levels of autophagy markers in hepatic stellate cells[34], we found that Nogo-B interacts with ATG5 to promote autophagy flux, which leads to LD breakdown in oxLDL-loaded HCC cells. Considering the vital role of ER in LD dynamics, we further determined whether these functions are dependent on the location of Nogo-B. Both LD degradation analysis and co-immunoprecipitation assay indicated that ER-retention characteristic of Nogo-B is required for its promoted effect on lipophagy-mediated lipid metabolism. The interaction between ER-bound Nogo-B and ATG5 occurs in LDs that are formed at the ER membrane[30]. When LDs are degraded by lipophagy, the ATG5-containing phagophores would enwrap the LDs for clearance[21].

The role of autophagy in HCC is paradoxical. Basal autophagy acts as a tumor suppressor by removing damaged mitochondria and mutated cells, thus maintaining genomic stability. However, once a tumor is established, unbalanced autophagy will contribute to HCC cell survival under various stress conditions and in turn promotes tumor growth[35]. In support of this notion, the liver-specific Atg5-deficient mice developed only hepatic adenoma but not cancer[36]. In addition, the development of DEN-induced HCC was suppressed by Atg5-deficiency[37]. In our present study, hepatic Nogo-B knockdown consistently disrupted ATG5-associated lipophagy, resulting in reduced DEN-associated tumorigenicity. In NAFLD-associated HCC, the role of autophagy is more complex because of the complicated linkages of lipid metabolism, inflammation and tumorigenesis. Although mTORC1-regulated autophagy is reported to be necessary

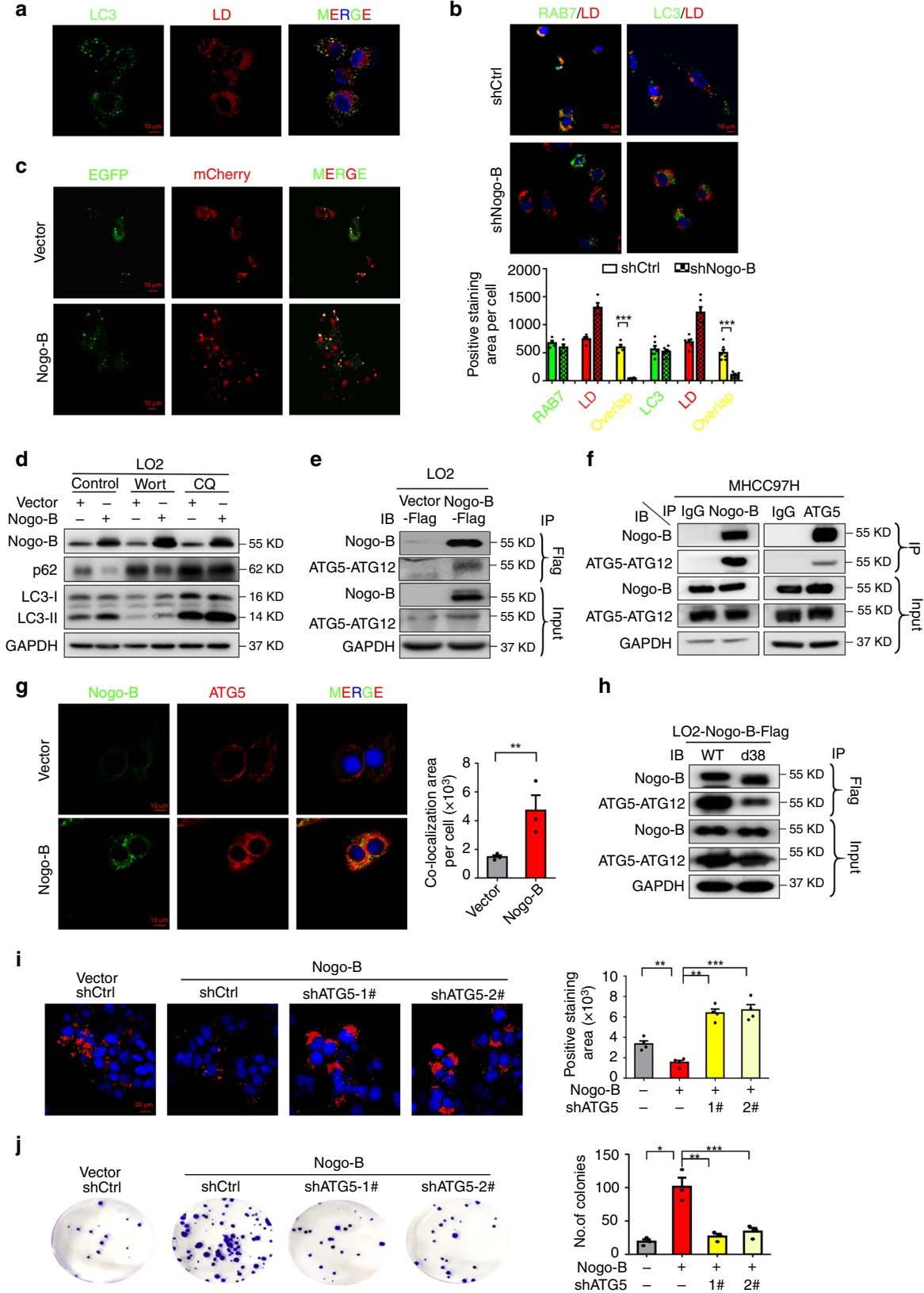

and sufficient for starvation-induced LD biogenesis, another study showed that the blockade of autophagy by pharmacological inhibitor or silencing ATG5 caused the reduction of LD and TG breakdown[7,38]. In our study, we revealed the lipolytic role of Nogo-B in the liver is dependent on autophagy-mediated LD degradation instead of LD formation. Our data supports

the role of autophagy in promoting lipid utilization and tumorigenesis.

Our results from qPCR Array of fatty liver-related genes indicates that CD36 is highly upregulated in tumors of HFHC-promoted HCC mice. As a fatty acid translocase, CD36 is widely expressed in various tissues and mainly involved in lipid

**Fig. 5** Nogo-B promotes lipophagy in HCC cells. **a** Immunofluorescence of LC3 and LD in LO2 cells treated with oxLDL for 24 h followed by starvation. **b** Representative immunofluorescence images (top) and quantification of co-localization areas (bottom) of RAB7 and LD, and LC3 and LD in control or Nogo-B-knockdown SK-Hep1 cells treated with oxLDL for 24 h followed by starvation. **c** Immunofluorescence of control and Nogo-B-overexpressing LO2 cells transfected with mCherry-EGFP-LC3 plasmid for 48 h followed by starvation. **d** Western blot analysis of Nogo-B, p62 and LC3 expression in starved control or Nogo-B-overexpressing LO2 cells treated with or without wortmannin (Wort) or chloroquine (CQ). **e, f** Co-immunoprecipitation of Nogo-B and ATG5 in **e** control or Nogo-B -overexpressing LO2 cells and **f** MHCC97H cells treated with oxLDL for 24 h followed by starvation. **g** Representative immunofluorescence images (left) and quantification of co-localization areas (right) of Nogo-B and ATG5 in starved control or Nogo-B-overexpressing LO2 cells. **h** Co-immunoprecipitation of Nogo-B and ATG5 in wile-type Nogo-B (WT) or ER-motif-deficient Nogo-B (d38) ectopically expressed LO2 cells treated with oxLDL for 24 h followed by starvation. **i** Representative LipidTOX staining (left) and quantification of positively stained areas (right) of control or Nogo-B-overexpressing LO2 cells transfected with ATG5 shRNAs for 48 h and treated with oxLDL for 24 h followed by starvation. **j** Representative image (left) and quantification of colonies (right) of control and stable Nogo-B-overexpressing LO2 cells transfected with ATG5 shRNAs for 2 weeks. (Data are presented as mean ± SEM of three independent experiments in **b**, **g**, and **i–j**. *$p < 0.05$, **$p < 0.01$, ***$p < 0.001$. Source data are provided as a Source Data file)

metabolism[39,40]. Although CD36 is implicated in NAFLD and HCC[40,41], the molecular action of CD36 in liver disease has remained poorly defined. We found a significant correlation between CD36 and Nogo-B in NAFLD-associated tumors from both murine models and clinical specimens. Based on the reported evidence that oxLDL was one of the major ligands of CD36 as well as the studies to validate the critical role of oxLDL in NAFLD development[13], we further observed a higher level of oxLDL that contributes to Nogo-B induction in NASH mice. Moreover, according to a study that showed upregulation of CEBPβ upon oxLDL exposure[15], we demonstrated that CEBPβ directly increases Nogo-B through promoter occupancy, which is required for oxLDL-stimulated Nogo-B expression. As hypoxia has been reported to increase lipid uptake and upregulate the expression of oxLDL receptor CD36, the increased Nogo-B levels could be a reflection of hypoxia in tumors[42]. Although the molecular mechanism by which the oxLDL-CD36 axis promotes CEBPβ expression remains to be defined, our findings underscore the regulatory role of the oxLDL-CD36-CEBPβ cascade to Nogo-B expression in NAFLD-associated hepatocarcinogenesis.

Cancer cells reprogram their metabolic pathways to meet their abnormal demands for proliferation and survival. Cancer cells are exposed to intermittent hypoxic episodes in the acidic microenvironment, which interferes with the metabolism of oxygenated cancer cells. Two recent studies demonstrated that cancer cells subjected to an acidic environment (pH 6.5) have a preferred glutamine reductive metabolism[43], and fatty acid oxidation to provide acetyl-coA to the tricarboxylic cycle. The consequences of this rewiring of metabolism are the preferred consumption of fatty acids to provide energy, but also the reduction of ROS production, which together support cancer cell proliferation and tumor growth[44]. In the absence of available nutrients, cancer cells can withstand long periods of nutrient deprivation via the self-catabolic process of macroautophagy to liberate free amino and fatty acids[45]. In fatty liver-related HCC, tumor cells tend to breakdown LD to provide fatty acids to the starved cells[17]. On the other hand, lipid metabolism in cancer cells also stimulates the common oncogenic signaling pathways, and is believed to be important for the initiation and progression of tumors[46,47]. In our study, by exploring the transcriptional regulation of Nogo-B, we identified oxLDL as a crucial lipid during NAFLD-associated tumorigenesis. We also elucidated that oxLDL-induced LD is degraded by Nogo-B-mediated lipophagy, which further leads to enhanced LPA level in HCC cells. Although autophagy has been demonstrated to be favorable for tumor growth and progression in advanced tumor stages[17,48], the role of lipophagy in cancer cells has not been identified. Nogo-B interacting with ATG5 participates in the fusion of phagophores and LD to breakdown the lipids. As tumor-secreted LPA is reported to aggravate HCC[49], the metabolic conversion of oxLDL to LPA that is mediated by lipophagy at least partially explains the oncogenicity of Nogo-B. Knockdown of *ATG5* abolished Nogo-B-induced LPA and YAP activity in vitro. Whether ATG5 is causally involved in Nogo-B-induced NAFLD-associated hepatocarcinogenesis warrants further investigation by in vivo rescue experiment. Recent reports have identified links between canonical Wnt signaling and the Hippo pathway, and several genes have been shown to take part in the crosstalk between the two pathways[50,51]. In our luciferase assay shown in Fig. 6a, in addition to the Hippo-YAP pathway, the transcription factor TCF4/LEF1 of Wnt/β-catenin signaling was also induced by Nogo-B overexpression, indicating that there may be other oncogenic mechanisms involved in Nogo-B stimulated metabolic dysregulation.

Increasing evidence supports the notion that oncoproteins directly reprogram the metabolism of tumor cells, which makes them excellent chemotherapeutic targets for cancer treatment. Our present study delineates the previously undiscovered function of Nogo-B in supporting autophagic flux, Hippo pathway suppression, and tumor cell proliferation. This autophagic function triggers Hippo pathway dysregulation via reprogrammed metabolic properties and requires Nogo-B-induced LD degradation. Hence, Nogo-B is a critical link between the enhanced oxLDL levels and the cancer cell metabolic reprogram that drives the development of NAFLD-associated HCC. Our discovery suggests that targeting Nogo-B may be an effective therapeutic strategy for tackling the increasing HCC incidence in metabolic syndrome patients.

## Methods

**Patients and clinical specimens**. Patients who underwent hepatectomy for NAFLD-associated HCC at the Prince of Wales Hospital (Hong Kong, China) were included in this study. All patients were defined as characteristic of metabolic syndrome, including diabetes, hypertension, dyslipidemia, or fatty liver changes, but no record of virus infection nor alcoholic intake (Supplementary Table 8). Studies using human specimen were approved by the joint CUHK-NTEC Clinical Research Ethics Committee. The patients were informed, and they signed consent forms acknowledging the use of their resected tissues for research purposes. The DNA copy number data were obtained from the cBioportal and Oncomine websites. The data used for screening of ER network gene expression and correlation analysis of mRNA and DNA copy number were obtained from TCGA database. The survival data were retrieved from the cBioportal website.

**Quantitative expression profiling**. For ER-residential genes expression profiling, total RNAs extracted from three pairs of tumor and adjacent non-tumor tissues of NASH-associated HCC mice were used for expression profiling of 103 ER-residential genes (Supplementary Table 1) by customized PCR Array. These 103 genes expressions were also detected in 360 HCC samples and 50 adjacent non-tumor liver tissues from HCC patients from TCGA database. The common significant differentially expressed genes were then verified by qRT-PCR and western blot analysis using additional 12 pairs tumor and non-tumor tissues from HFHC-fed mice. For fatty liver genes expression profiling, total RNAs extracted from four pairs of tumor and adjacent non-tumor tissues of NASH-associated HCC mice were used for expression profiling of 85 fatty liver genes (Supplementary Table 2) by PCR Array (Qiagen, PAHS-157Z).

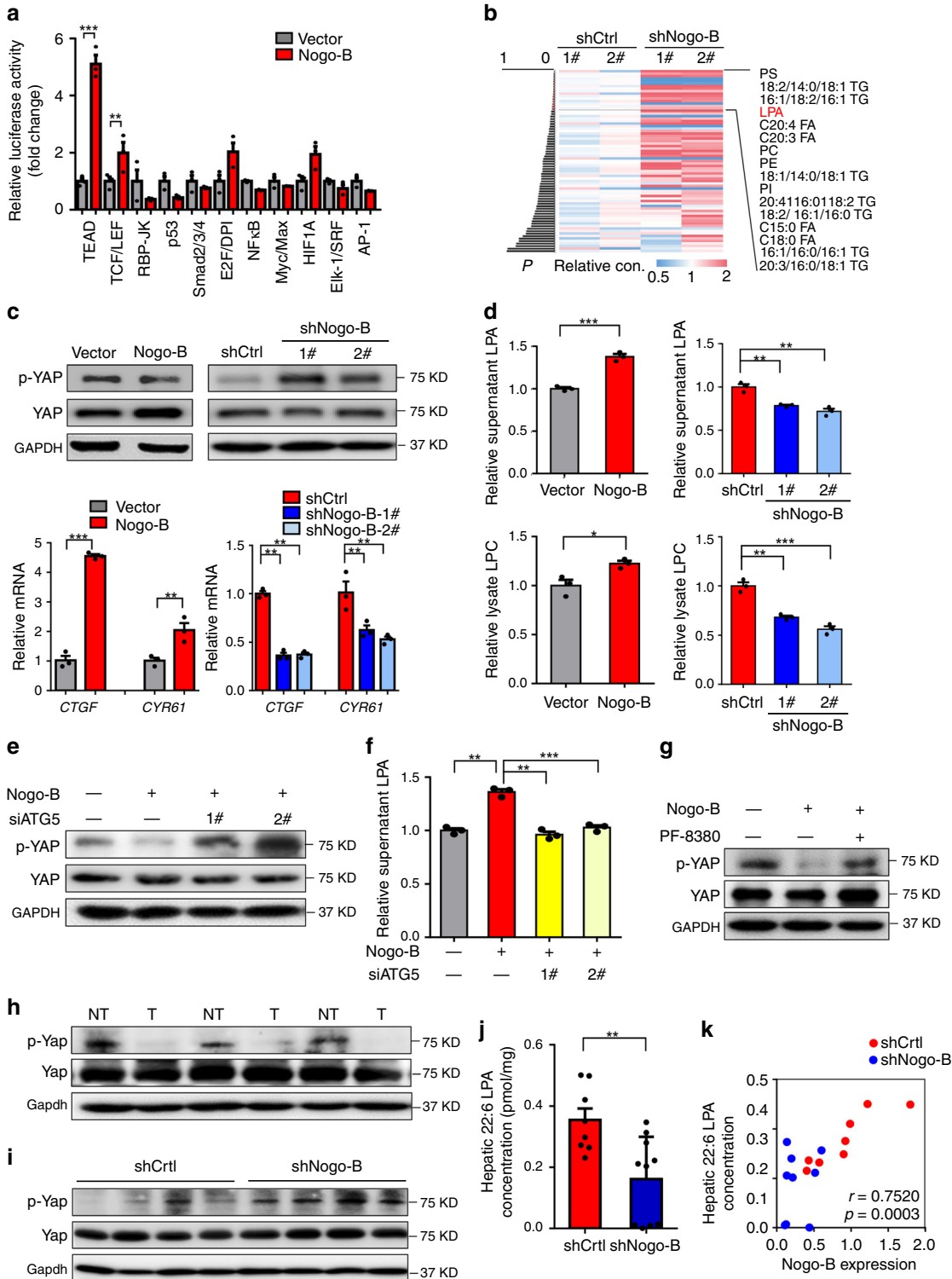

**Cell culture**. Hep3B, HepG2, PLC5 SK-Hep1, and RAW264.7 cells were obtained from the American Type Culture Collection (ATCC, Manassas, Virginia, USA). Huh7 cells were obtained from the Japanese Collection of Research Bioresources (JCRB, Tokyo, Japan). LO2 and BEL-7404 were obtained from the Cellosaurus. SMMC-7721 cell line was purchased from the Shanghai Cell Bank Type Culture Collection Committee (CBTCCC, Shanghai, China). MHCC97L and MHCC97H cells were gifts from Fudan University (Dr. Zhaoyou Tang) of Shanghai. HepG2.2.15 cell line was a gift from Eastern Hepatobiliary Surgery Hospital (Dr. Hongyang Wang) of Shanghai[52]. All cells were maintained in high-glucose Dulbecco's Modified Eagle Medium (DMEM, Gibco) supplemented with 10% fatal bovine serun (FBS, Hyclone). LO2 cell line was maintained in high-glucose DMEM (Gibco) supplemented with 10% FBS (Hyclone) and MEM Non-Essential Amino

Acids solution (Hyclone). The cells were incubated at 37 °C in a humidified chamber containing 5% $CO_2$. Cell lines were authenticated by short tandem repeats (STR) profiling and confirmed to be mycoplasma free.

**RNA interference and transfection**. Cells were transfected with 100 nM siRNAs (Supplementary Table 4) against Nogo-B and a control sequence using lipofecta-mine 2000 (Invitrogen) according to the manufacturer's protocols. The expression vectors were transfected into cells using viafect (Promega) according to the man-ufacturer's instructions. Stable transfectants were selected for 4 weeks with appropriate antibiotics. Resistant colonies were isolated, and individual clones were expanded in the selection medium.

**Fig. 6** Nogo-B stimulates the Hippo pathway in HCC cells. **a** Pathway reporter luciferase array revealed signaling deregulation by Nogo-B in LO2 cells. Cells were transfected with different pathway luciferase reporters for 48 h. **b** Heatmap of lipidomics signatures in livers from HFHC-promoted HCCs ($n = 2$ in each group). **c** Western blot (top) and qRT-PCR (bottom) analyses of YAP activity in LO2 cells upon ectopic expression (left) and MHCC97H cells upon knockdown (right) of Nogo-B. **d** LPA concentrations in the supernatant (top) and LPC concentrations in the cell lysate (bottom) of LO2 cells upon Nogo-B ectopic expression (left) and MHCC97H cells upon Nogo-B knockdown (right). Stable cell lines were treated with oxLDL for 24 h followed by starvation. Supernatant and cell lysate were collected for detection. **e** Western blot analysis of p-YAP and YAP in control or Nogo-B-overexpressing LO2 cells transfected with ATG5 siRNAs for 48 h and treated with oxLDL for 24 h followed by starvation. **f** LPA concentrations in the supernatant of control or Nogo-B-overexpressing LO2 cells transfected with ATG5 siRNAs for 48 h and treated with oxLDL for 24 h followed by starvation. **g** Western blot analysis of p-YAP and YAP in control or Nogo-B-overexpressing LO2 cells with treatment of oxLDL for 6 h and then additional PF-8380 (250 nM) treatment for 36 h followed by starvation. **h** Western blot analysis of p-Yap and Yap in paired tumors (T) and normal tissues (NT) from HFHC-fed mice ($n = 3$ in each group). **i** Western blot analysis of p-Yap and Yap of the liver tissues from HFHC-fed mice administered with lentivirus expressing shCtrl or shNogo-B ($n = 4$ in each group). **j** 22:6 LPA concentrations in the liver tissues from HFHC-fed mice administered with lentivirus expressing shCtrl ($n = 8$) or shNogo-B ($n = 10$). **k** Correlation analysis of 22:6 LPA concentrations and Nogo-B expression levels in the liver tissues from HFHC-fed mice administered with lentivirus expressing shCtrl ($n = 8$) or shNogo-B ($n = 10$). (Data are presented as mean ± SEM of three independent experiments in **a**, **c**, **d**, and **f**. *$p < 0.05$, **$p < 0.01$, ***$p < 0.001$. Source data are provided as a Source Data file)

**Quantitative RT-PCR (qRT-PCR).** Total RNA was extracted by using TRIzol reagent (Invitrogen). Five-hundred nanograms of RNA was reverse transcribed to cDNA using Reverse Transcription Master Kit (TAKARA) according to the manufacturer's instructions. For quantitative PCR analysis, aliquots of cDNA were amplified using Power SYBR Green PCR Master Mix (TAKARA) and ViiA7 Real-Time PCR System (Applied Biosystems). GAPDH was used as an internal control. All reactions were performed in triplicate. The PCR primers are listed in Supplementary Table 5.

**Western blot.** Protein lysates from cell lines and tissues were prepared using protease inhibitor cocktail (Roche) containing lysis buffer (50 mM Tris-HCl, pH 7.5, 150 mM NaCl, 1% NP-40, 0.5% Na-deoxycholate), and T-PER Tissue Protein Extraction Reagent (Thermo Scientific), respectively. Protein concentration was determined by the Bradford method (Bio-Rad Laboratories). In all, 20–100 μg of protein was resolved by 12% SDS-polyacrylamide gel electrophoresis and electro-blotted onto equilibrated nitrocellulose membrane (Bio-Rad Laboratories). Membranes were incubated with primary antibodies at 4 °C overnight followed by secondary antibodies for 2 h at room temperature. Information on the antibodies are provided in Supplementary Table 6. Antibody–antigen complexes were detected using the Western Blotting Chemiluminescence Luminol Reagent (Millipore). Signals from tissues were quantified by Imagine J software and defined as the ratio of target protein relative to GAPDH. The uncropped blots are shown in Supplementary Fig. 8.

**Cell growth assay.** Eight-hundred cells seeded on 96-well plate were transiently transfected with plasmids (Supplementary Table 7). The cell numbers were determined by MTS assay every 24 h for 4 consecutive days after transfection. All experiments were performed in triplicate.

**Colony formation assay.** Twenty-thousand cells seeded on 12-well plate with 50 to 80% confluence were transfected with vectors or infected with lentivirus. After 2 days, 200 cells were reseeded onto a 6-well plate and cultured for 2 weeks in antibiotic containing selection medium. The resistant colonies were stained with 0.2% crystal violet and counted under the microscope. Data were obtained from three independent experiments.

**Lipid droplet degradation and formation assay.** For LD degradation assay, cells were seeded on 12-well-plate with 30% confluence and then pretreated with oxLDL (80 μg/mL) for 24 h followed by starvation in 0.1% FBS for 12 h. For LD formation assay, cells were treated with oxLDL (80 μg/mL) for 6 h. Cells were fixed in 4% formaldehyde for 10 min at room temperature. After rinsed with phosphate buffer saline (PBS) for three times, the cells were stained with LipidTOX (1:1000, Invitrogen) for 30 min and counterstained with DAPI (Invitrogen) according to manufacturer's protocol. Images were captured using confocal microscope (Zeiss LSM 700).

**Orthotopic mouse model.** An orthotopic HCC mouse model was used to determine intrahepatic tumorigenicity. Briefly, $5 \times 10^6$ cells were injected subcutaneously into the dorsal right flank of female athymic nude mice. Subcutaneous tumors were harvested 4 weeks after injection and cut into 1.0 mm³ pieces. One piece was then implanted into the left liver lobe of each mouse. The mice were sacrificed after 5 weeks, and the tumor size and weight were measured. All experiments were performed in accordance with relevant institutional and national guidelines and the experimental procedures on use and care of animals had been approved by the ethics committee of Institute of Biophysics, Chinese Academy of Sciences.

**Xenograft mouse model.** Studies using female athymic nude mice (4- to 6-weeks-old) were reviewed. In all, $5 \times 10^6$ cells were subcutaneously injected into the right and left flanks of the mice. Tumor size was measured every other day using a caliper, and the tumor volume was calculated as $0.5 \times l \times w^2$, with $l$ indicating length and $w$ indicating width. The mice were euthanized at 4–6 weeks, and the tumors were excised and snap-frozen.

**NASH-associated HCC mouse model.** DEN (25 mg/kg, Sigma-Aldrich) was injected i.p. into 14-day-old C57BL/6 mice. After 6 weeks, mice were randomly assigned to LFD or high-fat, high-carbohydrate (HFHC) diet (Surwit diet) and drinking water enriched with high-fructose corn syrup. Lentiviruses encoding shRNA against Nogo-B (shNogo-B) or control sequence (shCtrl) were packaged according to the manufacturer's instructions (Dharmacon) for transduction in the dietary obesity models. At the age of 6 and 18 weeks, $5 \times 10^7$ transducing units of lentiviruses in 100 mL PBS were administered via tail vein injection. IPGTT was performed at the age of 26 weeks. All mice were sacrificed when 28 weeks old, and liver and blood samples were collected for expression analysis and metabolic profiling[53].

**Intraperitoneal glucose and insulin tolerance tests (IPGTT).** Twenty-six-week-old mice were fasted overnight for ~16 h and transferred to new clean cages. The next day, 1.5 g glucose per kg body weight was administrated intraperitoneally. Tail blood was then taken under normal condition at specific time-points. Glucose levels were determined using blood glucose strips (Johnson & Johnson).

**Metabolic profiling.** Heart blood was collected from 28-week-old mice before sacrifice and kept at room temperature for no more than 4 h. Centrifuge the blood at 3000 rpm at 4 °C for 15 min and transferred the serum to new tubes. Store the serum at –20 °C before use. The concentration of serum FFA, TG, oxLDL, and LDL were determined using respective ELISA kits (FFA, TG: LabAssay; oxLDL, LDL: Cloud-Clone) according to the manufacturer's instructions.

**Lipid droplet isolation.** The liver from fasted C57B/L6 mouse was collected into ice-cold PBS containing 0.2 mM PMSF and transferred to 12 mL buffer A (25 mM tricine pH 7.6, 250 mM sucrose) plus 0.2 mM PMSF, homogenized with a Dounce type glass-Teflon homogenizer on ice. The homogenated tissues were centrifuged at $100 \times g$ to remove the debris and the cell suspension was further homogenized by N2 bomb (500 psi for 15 min on ice). The postnuclear supernatant fraction (10 mL) with 2 mL buffer B (20 mM HEPES, pH 7.4, 100 mM KCl, and 2 mM MgCl₂) was obtained by centrifugation at $3000 \times g$ and loaded into a SW40 tube. The sample was centrifuged at 38,000 rpm for 1 h at 4 °C. The white band containing lipid droplets at the top of gradient was collected in a 0.5 mL tube, then the sample was centrifuged at $20,000 \times g$ for 3 min, the underlying solution and pellet were carefully removed and discarded using a gel-loading tip, and droplets were gently resus-pended in 200 μL of buffer B for the further repeated washing and centrifugation as mentioned above. And finally we took the upper layer (white band) after cen-trifugation for western blot analysis of lipid droplets.

**Quantitative ChIP-PCR assays.** For target gene validation, PCR primers targeting a region within 150 base-pairs of the putative binding site were designed to detect IP and input DNA. IP and 2% input DNA were used as a template for conventional PCR assay. For quantitative ChIP-PCR, equal amounts of IP and diluted input DNA were used for Power SYBR Green-based detection (Applied Biosystems). The sequences of primers used are listed in Supplementary Table 2.

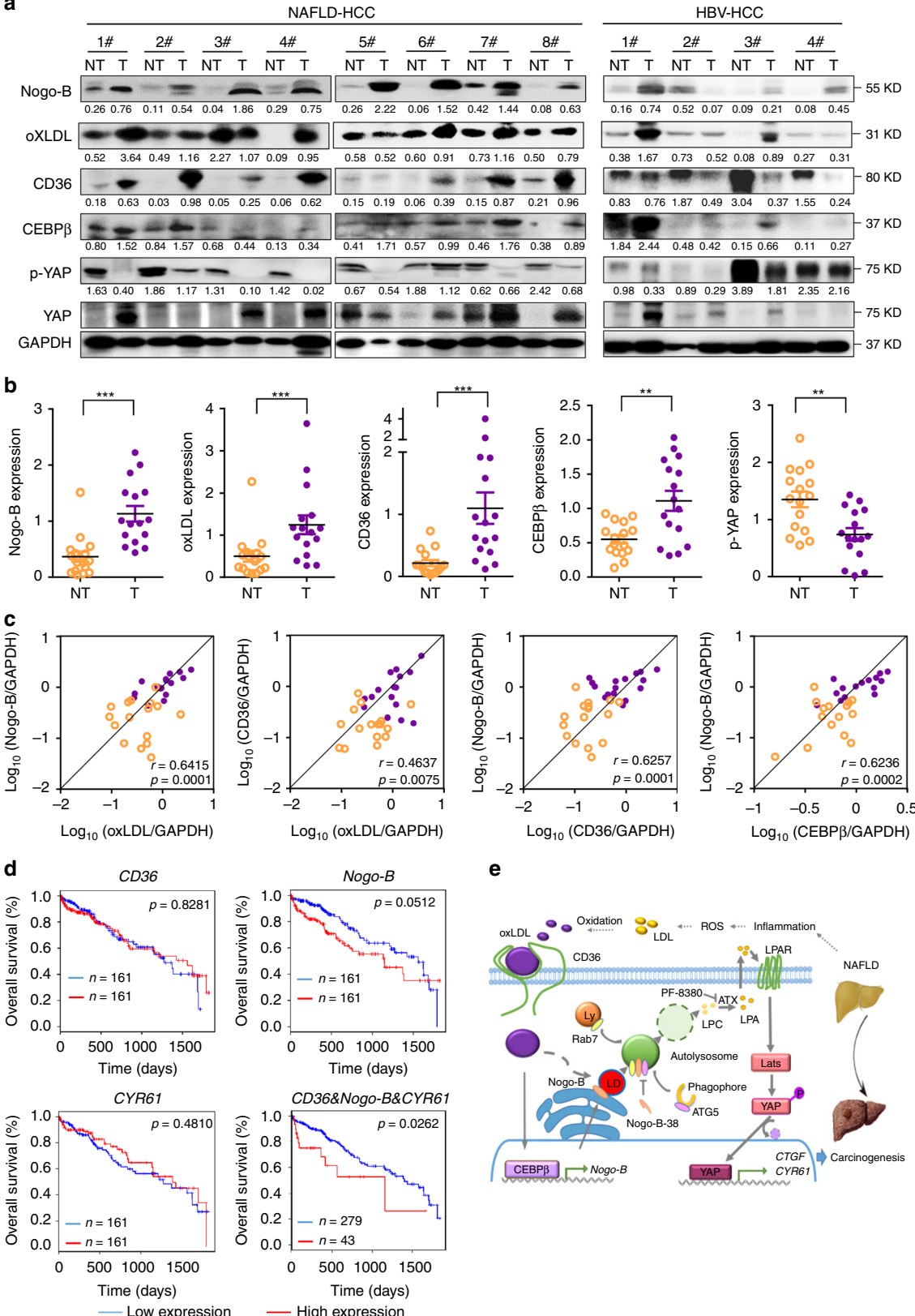

**Fig. 7** Enhanced Nogo-B cascade in clinical NAFLD-associated HCCs. **a** Representative blots of western blot analysis of upregulation of Nogo-B, oxLDL, CD36, and CEBPβ, and inhibition of YAP phosphorylation in eight paired human NAFLD-associated HCCs (left) and four paired human HBV-associated HCCs (right) (T, tumors; NT, adjacent normal tissues). **b** Quantification of Nogo-B, oxLDL, CD36, CEBPβ, and p-YAP in 16 pairs of NAFLD-associated HCC patient samples. (*$p < 0.05$; **$p < 0.01$; ***$p < 0.001$. Source data are provided as a Source Data file). **c** Correlation analysis of protein levels of Nogo-B and oxLDL-CD36-CEBPβ cascade in 16 pairs of NAFLD-associated HCC patient samples. **d** Kaplan–Meier analyses for HCC patients with tumors expressing high or low level of CD36, Nogo-B, or CYR61. **e** Working model of Nogo-B-promoted LD degradation and activation of YAP signaling in NAFLD-associated HCC

**Luciferase reporter assay**. LO2 cells were transiently transfected using costumed Cancer Pathway Reporter Arrays according to the manufacturer's protocols (SABiosciences). Cells were harvested 48 h after transcription and assayed by the Dual Luciferase Reporter Assay System (Promega) using GloMax microplate luminometer (Promega). The TEAD binding domain (TBD) luciferase plasmids and Renilla plasmid was kindly provided by Professor Faxing Yu.

**Immunohistochemistry**. Five-millimeter sections from formalin-fixed paraffin embedded archive tissues were deparaffinized, rehydrated, and rinsed in distilled water. Antigen retrieval was done by using a pressure cooker with 1 mM EDTA buffer, pH 8.0, for 10 min. The endogenous peroxidase activity was then blocked by incubating the slides in 3% hydrogen peroxide in methanol for 30 min. The sections were then stained with polyclonal antibody against Nogo-B (1:50, R&D) at 4 °C for 16 h, and chromogen development was performed using the universal HRP Multimer Ultraview Kit on Benchmark XL (Ventana Medical System).

**Immunofluorescence**. Cells grown on coverslips were fixed with 3% paraformaldehyde and permeated with 0.1% Triton X-100. Nonspecific binding was blocked with 1% BSA for 30 min. The cells were then incubated with primary antibodies against rabbit anti-Nogo-B at 4 °C overnight, followed by rhodamine-conjugated goat anti-rabbit antibody (Invitrogen) for 30 min. The cells were then blocked with 1% BSA for 30 min followed by incubating with primary antibodies against rabbit anti-LC3 (Sigma-Aldrich) at 4 °C overnight, followed by rhodamine-conjugated goat anti-rabbit antibody (Invitrogen) for 30 min. Nuclei were counterstained by DAPI (Invitrogen). Images were captured using confocal microscope (Zeiss LSM 700 & Zeiss LSM 880).

**Co-immunoprecipitation**. BEL-7404 and MHCC97H cells were collected and lysed in 1 ml co-IP lysis buffer (20 mM Tris, pH 7.5, 150 mM NaCl, 1.0% Triton X-100, 1 mM EDTA, and protease inhibitor cocktail) on ice. The lysates were centrifuged at $12,000 \times g$, 4 °C, 15 min and immunoprecipitated with IgG or Nogo-B antibodies at 4 °C overnight. The next day, 20 μL protein G beads (Santa Cruz) were added into the protein–antibody complex and incubated for 6 h at 4 °C. LO2 cells were lysed and immunoprecipitated with Flag-beads overnight. The complexes were then rinsed with Co-IP lysis buffer four times for 10 min each at 4 °C and collected by centrifuging at 2500 rpm, 4 °C, 5 min. Proteins were detached from beads by denaturing with 4 × protein loading buffer at 100 °C for 10 min on a heat block (Thermo Fisher Scientific). Protein interaction was then detected by western blot using respective antibodies.

**Lipidomics and LPA concentration assay**. For cell lysate, LPA and LPC levels were measured by ELISA kit (Cloud-Clone, CEK623Ge and CEK621Ge) according to the manufacturer's instructions. For tissues used in lipidomics and LPA analyses, frozen liver tissues were crushed and transferred into glass tubes, followed by adding water to make a final concentration of 200 mg/mL. Tissue powders were homogenized using a Brinkmann POLYTRON PT 10/35 Homogenizer and 50 μL sample was transferred into 450 μL methanol. After vortexing and centrifugation ($10,000 \times g$, 10 min, 4 °C), 200 μL of the supernatant was used for high-performance liquid chromatography mass spectrometry (HPLC/MS) analysis. Mass analyses were performed on the API-4000 mass spectrometer (Applied Biosystems/MDS SCIEX, Forster City, CA) with the Analyst data acquisition system. Negative ion MRM mode was used for the quantitative analysis of LPAs.

**Separation of cell subcellular fractions**. Nuclear and cytoplasmic protein was separated by using Nuclear and Cytoplasmic Protein Extraction Kit (Beyotime, P0027) according to the manufacturer's instructions. The cytoplasmic extract and nuclear extract were used for western blot analysis. The Isolation of the endoplasmic reticulum (ER) from cultured cells was performed by using the ER Isolation Kit (Sigma, ER0100) accordingly to the manufacturer's instructions. Briefly, the post mitochondrial fraction (PMF) was obtained according to the procedure. Then PMF was precipitated with calcium chloride to isolate rough ER (RER) enriched microsomes according to the procedure Option 1. All procedures were performed at 4 °C.

**Transmission electron microscopy**. Cells were collected and fixed with 2.5% (vol/vol) glutaraldehyde with phosphate buffer (PB) (0.1 M, pH 7.4), washed four times in PB. Then cells were postfixed with 1% (wt/vol) osmium tetroxide in PB for 2 h at 4 °C, dehydrated through a graded ethanol series (30, 50, 70, 80, 90, 100%, 100%, 7 min each) into pure acetone (2 × 10 min). Samples were infiltrated in a graded mixtures (3:1, 1:1, 1:3) of acetone and SPI-PON812 resin (16.2 g SPI-PON812, 10 g DDSA and 8.9 g NMA), then pure resin was changed. Finally, cells were embedded in pure resin with 1.5% BDMA and polymerized for 12 h at 45 °C, 48 h at 60 °C. The ultrathin sections (70 nm thick) were sectioned with microtome (Leica EM UC6), double-stained by uranyl acetate and lead citrate, and examined by a transmission electron microscope (FEI Tecnai Spirit120kV).

**Statistics**. Unless otherwise indicated, data are presented as mean ± SEM of three independent experiments. Sample sizes were chosen to satisfy statistical power

based on previous experience and knowledge. For animal studies, we performed power analyses using a web-based tool at www.biomath.info. Blinding in in vivo experiments was not done during experimentation, but the labels were covered during data analysis. GraphPad Prism 5 was used for data analysis. The independent Student's $t$-test was used to test the significance of differences between two selected groups. The protein levels in the HCCs and the matched non-tumor tissues were compared using nonparametric Wilcoxon's matched pairs test. The correlation between protein expressions was analyzed using Pearson's test. A two-tailed $p$-value< 0.05 was considered statistically significant. The Cox univariate proportional hazards regression model was used to determine the independent clinical factors based on the investigated variables. R function cox.zph was used to test the proportionality assumption. The confidence interval for each variable, $p$-values and hazard ratios are shown (Supplementary Tables 9–11).

**Reporting summary**. Further information on research design is available in the Nature Research Reporting Summary linked to this article.

## Data availability
The TCGA data reference in this study are available in the cBioPortal for Cancer Genomics website and Oncomine website. The authors declare that all the other data supporting the findings of this study are available within the article and its Supplementary Information files and from the corresponding author upon reasonable request. The source data underlying all bar charts are provided as a Source Data file.

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

## Acknowledgements

We thank Prof. Pingsheng Liu, Prof. Xiaochen Wang, Prof. Hong Zhang, Dr. Shuyan Zhang, Dr. Hongyu Zhao, Dr. Guangyan Miao, and Mr. Nan Liu (IBP, CAS) for their advice and technical support. We also thank Can Peng, and Lei Sun for their help to use electron microscopy at the Center for Biological Imaging (CBI at IBP, CAS). The plasmid pGL4.20-4 × TBD was kindly provided by Prof. Faxing Yu (Fudan University). The Atg5 KO MEFs and ATG7 antibody were kindly provided by Prof. Hong Zhang (IBP, CAS). This work was supported by grants: the Strategic Priority Research Program of the Chinese Academy of Sciences (No. XDB29040102), National Key Basic Research Program of China (No. 2015CB553705 ), the Strategic Priority Research Program of the Chinese Academy of Sciences (No. XDA01020304), NSFC (No. 81522030, 81672464, 81602453, 81672464, 91439132, 81472282), the University Grants Committee through the Collaborative Research Fund C4045-18W, General Research Fund 14120816 and the Focused Innovations Scheme-Scheme B 1907309 from the Chinese University of Hong Kong and the Li Ka Shing Foundation (Canada).

## Author contributions

P.Y. initiated and supervised the project; Y.T. and P.Y. designed the experiments; Y.T., B.Y., W.Q., Z.Z., B.Y. and Z.L. performed the experiments, collected and analyzed data; Y.H., O.K.C., J.L. and R.C. helped bioinformatics analysis; N.L., S.C., Y.W., Y.S.C., P.B.L., H.Y.W., J.J.S. and A.S.C. acquired and managed the clinical samples; W.K.W. and A.S.C. helped on the paper revision; Y.T. and P.Y. wrote and reviewed the manuscript.

## Additional information

**Competing interests:** The authors declare no competing interests.

