## [Peer Review File · Nature Communications]

Reviewers' Comments:

Reviewer #1:

Remarks to the Author:

In the current manuscript the authors identified Nogo-B as an accelerator of diet-induced metabolic dysfunction and tumor progression. Furthermore, they have demonstrated that Nogo-B which is regulated by cellular oxLDL uptake, interacts with ATG5 to promote lipophagy leading to lysophosphatidic acid-enhanced YAP oncogenic activity.

While several mechanistic aspects of the paper are novel, the main conclusion that Nogo-B is expressed in HCCs and plays an oncogenic role as been shown before (Zhu B et al, Neoplasia 2017). Nevertheless this paper provides important mechanistic studies that further explain the link between oxLDL and HCC.

Main comments:

- Regarding the association between Nogo-B and HCC: The increased expression of Nogo-B in tumors compared to paired adjacent normal tissues from the mice could be a reflection of the lack of oxygen around the tumor as hypoxia increases LDL oxidation and Nogo-B was already shown to be regulated by oxLDL. This point can be addressed in the discussion. Furthermore, details about the three HCC cohorts are missing; do all patients have NAFLD? What is the number of patients per cohort? Was the healthy tissues taken from the same individuals? Likewise, information about the healthy livers (figure 1D) is missing.
- Regarding the contribution of Nogo-B to tumor growth in NAFLD-associated HCC: In the comparison between Nogo-B stably transfected cells (Nogo-B) or empty vector transfected cells (vector), it is surprising to see no variation at all in the vector transfected cells. Was that indeed done in 5 mice? Also, not clear what is the small green line above week 23.
- Regarding the observations that Nogo-B is controlled by oxLDL-CD36-CEBP β cascade: CD36 is not the only receptor for oxLDL. What about other receptors (i.e. SRA?). Also, what about other ligand for oxLDL (i.e. fatty acids)? Is PPAR gamma also activated? Also, was there also an effect on CD36 translocation?
- Regarding the functional role of Nogo-B in NASH-associated carcinogenesis: the effect on inflammation should also be investigated (i.e. IL 6 expression).
- Regarding the interesting finding that Nogo-B promotes lipid droplet degradation: NAFLD is associated with high levels of FFAs compared to healthy liver. Why would additional FFAs from LD breakdown be essential for tumor growth under these conditions? This issue can be addressed in the discussion.

Minor comments:

- Immunostaining pictures should be scored.
- Manuscript should be checked for English.

Reviewer #2:

Remarks to the Author:

Authors investigated the role of Nogo-B, an endoplasmic reticulum resident protein, in non-alcoholic fatty liver disease (NAFLD)-associated hepatocellular carcinoma. Using a NAFLD associated HCC mouse model (mice injected with the carcinogen DEN followed by feeding with a high fat and high carbohydrate diet), authors identified Nogo-B as an associate factor which

promoted HCC development in mice. They further identified that increased oxLDL uptake via increased CD36 activated transcription factor CEBP β , which may further regulate the upregulation of Nogo-B. Furthermore, they found that Nogo-B localized on lipid droplet (LD) and physically interacted with Atg5. Overexpression of Nogo-B decreased LD numbers whereas the knockdown of Nogo-B increased LD numbers. Knockdown of Nogo-B in mouse livers also led to decreased liver tumorigenesis in NASH-HCC mouse model. Lipidomic studies revealed increased lysophosphatic acid (LPA) in NASH-HCC model and was associated with YAP activation. Authors concluded that increased LPA and YAP activation was due to Nogo-B induced lipophagy. Finally, authors were able to show Nogo-B expression positively correlated with NAFLD-associated HCC in human patients. There were many strengths of this manuscript. The study focused on NAFLD-associated HCC, which is a very important health problem currently. The study was quite comprehensive including in vitro, in vivo and human samples with robust transcriptome and lipidomic studies. The link of Nogo-B to lipophagy and YAP activation is also novel. The manuscript was well written and easy to follow.

However, there were also severe concerns on this manuscript in terms of data rigorous, weak autophagy and lipophagy data and the overlook of the complex role autophagy in liver tumorigenesis resulting in over interpretation of some of the data. There was also an insufficient discussion on the conflicting reports in the literature.

Major concerns:

1. The role of autophagy in liver tumorigenesis is quite complex. It is generally thought that at the early phase that autophagy protects against liver tumorigenesis, which is supported by the evidence that several liver specific Atg-deficient mice (atg5 or atg7) develop spontaneous liver tumor. However, these tumors are benign adenoma and autophagy actually is required at the late stage for the benign adenoma to progress to malignant carcinoma (Li K et al. Mol Cell 2017). The current manuscript failed to address this important issue. For instance, knockdown of Nogo-B in mice led to decreased NAFLD-associated HCC. If the Nogo-B mediated lipophagy hypothesis was correct, how the authors would reconcile these data with the literature. What was the nature of the liver tumors in the Nogo-B knockdown mice? What would happen if Atg5 be knocked down in this mouse model? This is important because it raised the possibilities that Nogo-B may use different mechanisms rather than autophagy/lipophagy to promote liver tumorigenesis.
2. Another major issue was the lack of experimental details in the figure legends in most figures or panels. It was unclear how many repeats were done and how many mice were used in some figures, which raised concerns on the rigor of the data and made it hard to evaluate.
3. The overall autophagy and lipophagy data were weak. Figure 5C need to be quantified. Only LC3-II levels were not sufficient to conclude autophagy activity. Figure 5D needs to add lysosomal inhibitors to assess the changes of LC3-II and p62. What was the molecular weight of ATG5 in figure 5? Was this the monomer of ATG5 or the conjugated form?
4. Figure 4, authors claimed increased LD staining after knockdown of Nogo-B in the starvation condition was due to the impaired lipophagy. However, it has been reported that knockdown of Atg genes or pharmacological inhibition of autophagy led to decreased LD numbers (Nguyen TB et al., Dev Cell 2017)? These controversial findings have not been discussed. Moreover, the area of lipid staining may not be the best way to quantify the lipid changes, the total cellular triglycerides should be used.
5. Another major pathway to mediate LD lipolysis was the triglyceride lipase (ATGL) and hormone sensitive lipase (HSL). Does Nogo-B also affect ATGL and HSL?
6. Figure 4C, what was the lysate FFA and TG? In general, lipid should be extracted for the measurement of FFA and TG. This should be better clarified.

Reviewer #3:

Remarks to the Author:

The manuscript from Cheng and Yang groups delineates the role of an ER-resident protein Nogo-B in NAFLD-associated HCC. The authors propose the link between elevated Nogo-B levels in tumors to high lipophagy and tumor cell proliferation. The data showing inverse relationship between

Nogo-B and HCC are striking and quite robust; however, the link between Nogo-B and lipophagy is weak and warrants further investigation.

Nogo-2 overexpression does not seem to increase autophagy flux as shown by the dual LC3 reporter. The number of red only dots (indicative of autolysosomes) seems to be similar to control in Figure S5B. Additionally, the authors show one cell for each group. Without quantification and statistics, the authors cannot make the claim that the differences are significant.

The images in Figure S5D are of poor quality. It would seem that there is 100% overlap between Nogo-B and ATG5, which seems difficult to believe and is more likely artifactual. Also, Nogo-B is an ER protein and the authors show it to be present on LD. The authors also show the association of autophagy proteins with Nogo-B. Now, is this interaction of Nogo-B occurring in the ER or at the LD? Is this an LD-ER contact site? Or is this simply ER-phagy and the lipids are inside the ER? The IP with the ER-motif mutant Nogo-B suggests the last option; however, additional experiments will be required to strengthen this connection.

Does Nogo-B have LIR-motifs? It would be interesting to pursue this line if the authors want to make a stronger case of its link with autophagy.

An autophagosome does not have ATG5 on it. So the depiction in the proposed working model (Figure 7E) is incorrect.

Does Nogo-B induce the expression of autophagy genes – ATG5 and LC3?

Instead of colocalizing Rab7 with LD, it would be more convincing to perform its colocalization with LC3 and LD in shNogo-B (Figure 5B), given the several non-autophagy roles of Rab7 and presence of Rab7 on other organelles.

Figure 2B: Is Nogo-B present in the gene array?

Presence of Nogo-B on LD is shown in normal mouse livers. Is the protein present on the LD of HCC and NAFLD models as well?

Nogo-B knockdown induced greater expression of LD marker ADFP (Figures S4B, S4C). However, no difference in the amount of LD was observed between shNogo-B and control cells in Figures 4E and 4F. Both legends say this was in response to oxLDL. This is an inconsistency.

Lysate FFAs are not a read-out of lipolysis. The cells generally don't have free fatty acids due to their toxicity.

The increase in TG upon KD of Nogo-B (Figure 4H) seems very modest. This does not match with the almost 3-fold increase in LD shown in Figure 4D even though the starvation time in Figure 4H is double that of Figure 4D (as per the legend).

Figure 1K: the images do not show a decrease in volume/weight.

Figure S4A: PLIN2 as LD marker is missing. ADFP should be replaced by PLIN2 as per current guidelines of naming perilipins/LD proteins.

Figure S4C: increased expression of PLIN2 (ADFP) in MHCC97H cells upon shNogo-B is not convincing.

Figure 4D: Resting shNogo-B control is missing.

For all IF images, please include a nuclear stain to bring some perspective for the readers. It is difficult to identify the cell/s, for example, in 5A.

Some of the immunoblots are pixelated and some of the others are over-exposed.

The authors use 'significantly' without providing quantification and statistics.

For LD isolation, the authors mention 3,300,000g centrifugation step with SW40 rotor. This is not possible. Please correct this error?

Most of the experiments with Nogo-B overexpression with Ctrl or sh ATG5 do not include the shATG5 only cells. In addition, the data showing that there is efficient KD of ATG5 with shRNA or siRNA is missing.

The details regarding the plasmids used in this study are missing.

The methodology for staining with LipidTOX is missing.

The methodology for treatment of cells with ox-LDL is missing.

How ox-LDL is measured in serum is missing.

There are several typographical errors in the manuscript that must to be corrected.

Reviewer #4:

Remarks to the Author:

In the article entitled "ER-Residential Nogo-B accelerates NAFLD-associated HCC mediated by metabolic reprogramming of oxLDL lipography" the authors aim to identify and understand the mechanism by which lipid metabolism is dysregulated in HCC related to fatty liver etiology. They identify the ER-protein Nogo-B, as a modulator of metabolism in NAFLD-HCC and players in the associated carcinogenic pathway. Given the importance of fatty-liver related disease, the study is of interest, however the manuscript would benefit from the following suggestions:

- 1) The initial array data shown in Suppl Fig 1A is not comprehensive as it only represents an average of 3 paired specimens. It would be useful to show each T and NT sample or each of the 3 ratio data to help determine the consistency of the finding. Is consistent data also seen in TCGA? The authors state so, but the data is not shown.
- 2) In Fig 1A, how many HCC specimens in TCGA are interrogated? Are the HCC specimens used in TCGA for this analysis in Fig 1A or Fig 1D and Suppl Fig 1C all NAFLD-related or are they from different etiologies? The authors show later in the manuscript that Nogo-B is altered in HBV-HCC. If the latter, then it is difficult to relate Nogo-B effects specifically to lipid metabolism.
- 3) The authors state that Nogo-B RNA copy number is upregulated in 3 cohorts but they do not show that copy number alterations are correlated or connected to expression changes of Nogo-B in the same samples. It is difficult to confirm its oncogenic function as the authors stet on line 106.
- 4) The authors screen several cell lines for Nogo-B level and identify two with lower levels and two with higher levels that are utilized for several experiments in the manuscript. However, in several cases, functional assays are only performed in one-representative cell line (e.g. Supp Fig 4E, 4G, 4H Fig 5B, H and I, 6C and D) and in Fig 5E anf F, the BEL-7404 cell line is used without rationale for not using either of the Nogo-B cell lines used in the previous figures.
- 5) In Fig 6B, data from only 2 specimens in either the control of Nogo-B group are shown. A larger panel of specimens should be assayed to determine differences in relative lipidomic levels.
- 6) In Fig 7, data for only 8 and 4 pairs of NAFLD-HCC or HBV-HCC are shown, however 16 and 12 are noted in the manuscript text.
- 7) The association of the 3 genes with reduced patient survival has not been tested for confounding variables.

Point-to-point responses to Reviewers

To Reviewer 1:

We appreciate the reviewer's compliment on our *manuscript's novelty* that “*provides important mechanistic studies that further explain the link between oxLDL and HCC*”.

Major comments:

Question 1. *Regarding the association between Nogo-B and HCC: The increased expression of Nogo-B in tumors compared to paired adjacent normal tissues from the mice could be a reflection of the lack of oxygen around the tumor as hypoxia increases LDL oxidation and Nogo-B was already shown to be regulated by oxLDL. This point can be addressed in the discussion. Furthermore, details about the three HCC cohorts are missing; do all patients have NAFLD? What is the number of patients per cohort? Was the healthy tissues taken from the same individuals? Likewise, information about the healthy livers (figure 1D) is missing.*

Response: We thank the reviewer for pointing out the relationship between hypoxia, LDL oxidation, and Nogo-B, which has now been added to the discussion (lines 386-388). We have also added the numbers of HCC and healthy liver tissues in each cohort in new Supplemental Figure 1D-G. Information describing the HCC cohorts is now provided in the Methods section and in the Results (lines 98-99, 106-107 and 433-437). As only ~3-4% of the HCC patients have NAFLD, we have directly compared the *Nogo-B* expression between 16 and 20 pairs of NAFLD- and HBV-associated HCC tumor/non-tumor tissues, respectively. Quantitative RT-PCR analysis showed more significant upregulation of *Nogo-B* in NAFLD-associated HCC cases (Figure 2A).

Question 2. *Regarding the contribution of Nogo-B to tumor growth in NAFLD-associated HCC: In the comparison between Nogo-B stably transfected cells (Nogo-B) or empty vector transfected cells (vector), it is surprising to see no variation at all in the vector transfected cells. Was that indeed done in 5 mice? Also, not clear what is the small green line above week 23.*

Response: The vector transfected group indeed consists of 5 mice. The image of the small tumors with consistently small size is shown in new Supplemental Figure 1K. We have also provided a clearer graph in new Figure 1H.

Question 3. *Regarding the observations that Nogo-B is controlled by oxLDL-CD36-CEBP β cascade: CD36 is not the only receptor for oxLDL. What about other receptors (i.e. SRA?). Also, what about other ligand for oxLDL (i.e. fatty acids)?*

Is PPAR gamma also activated? Also, was there also an effect on CD36 translocation?

Response: Out of the 85 fatty liver-associated genes screened in the PCR array, 6 receptors were included, namely *Cd36*, *Adipor2*, *Adipor1*, *Ldlr*, *Insr* and *Lepr*. We found that *Cd36* was significantly upregulated in murine NAFLD-associated HCCs compared to non-tumorous livers, while *Ldlr*, *Insr* and *Lepr* were downregulated and the others unchanged. The array data have been added in new Supplemental Table 2. In addition, we evaluated the expression of other oxLDL receptors, including *Sra*, *Sr-b1*, *Srec* and *Lox1*, in murine NAFLD-associated HCCs, and observed no consistent alteration in HFHC mice and tumor tissues (new Supplemental Figure 2G). While increased free fatty acid levels were observed in the HFHC group (new Supplemental Figure 2B), *Nogo-B* expression was not affected by oleic acid treatment (new Supplemental Figure 2F), in contrast to the robust upregulation observed upon oxLDL treatment (new Supplemental Figure 2D-E). PPAR gamma was not affected in this murine model (new Supplemental Table 2). Notably, CD36 was found to be translocated to the cell membrane and co-localized with Dil-oxLDL upon oxLDL stimulation (new Supplemental Figure 2C) supporting this ligand-receptor interaction in this system.

Question 4. *Regarding the functional role of Nogo-B in NASH-associated carcinogenesis: the effect on inflammation should also be investigated (i.e. Il 6 expression).*

Response: We have performed additional expression analyses and demonstrated that the levels of *Il6* and *Tnf-a* were upregulated in HFHC groups and restored upon *Nogo-B* knockdown (new Supplemental Figure 3B), suggesting it plays a role in modulating inflammation.

Question 5. *Regarding the interesting finding that Nogo-B promotes lipid droplet degradation: NAFLD is associated with high levels of FFAs compared to healthy liver. Why would additional FFAs from LD breakdown be essential for tumor growth under these conditions? This issue can be addressed in the discussion.*

Response: As shown in previous reports, FFAs from LD breakdown may provide ATP and key metabolites essential for tumor growth¹. This important issue is now discussed in more details in lines 393-397.

Minor comments:

Question 6. *Immunostaining pictures should be scored.*

Response: The immunostaining signals have now been scored and depicted in new Figure 5B, 5G, Supplemental 2C, 4A, 4D, 5B, 5C, and 5E.

Question 7. *Manuscript should be checked for English.*

Response: The manuscript has been checked for English.

To Reviewer 2:

We appreciate the reviewer's compliment on the "*many strengths of this manuscript*" which is "*quite comprehensive including in vitro, in vivo and human samples with robust transcriptome and lipidomic studies*", "*novel...well written and easy to follow*". We do have strengthened the manuscript by providing more data on autophagy and lipophagy through additional experiments to tease out its complex role, and have provided more robust discussion of the literature and how our study fits into the current understanding of this disease.

Major concerns:

Question 1. *The role of autophagy in liver tumorigenesis is quite complex. It is generally thought that at the early phase that autophagy protects against liver tumorigenesis, which is supported by the evidence that several liver specific Atg-deficient mice (atg5 or atg7) develop spontaneous liver tumor. However, these tumors are benign adenoma and autophagy actually is required at the late stage for the benign adenoma to progress to malignant carcinoma (Li K et al. Mol Cell 2017). The current manuscript failed to address this important issue. For instance, knockdown of Nogo-B in mice led to decreased NAFLD-associated HCC. If the Nogo-B mediated lipohagy hypothesis was correct, how the authors would reconcile these data with the literature. What was the nature of the liver tumors in the Nogo-B knockdown mice? What would happen if Atg5 be knocked down in this mouse model? This is important because it raised the possibilities that Nogo-B may use different mechanisms rather than autophagy/lipophagy to promote liver tumorigenesis.*

Response: We thank the reviewer for pointing out this important issue of the complex role of autophagy in liver tumorigenesis, which is now discussed in detail (lines 359-366). Based on the histological examination of the current study (Figure 3D) and our previous study ², malignant carcinoma occurred in this NAFLD-associated HCC model. Here, we found that Nogo-B interacts with ATG5 to promote lipophagy, leading to lysophosphatidic acid (LPA)-enhanced YAP oncogenic activity. Knockdown of *ATG5* abolished Nogo-B-induced LPA and YAP activity *in vitro* (Figure 6E-G and new Supplemental Figure 6E). In addition, Atg5 expression was also upregulated in

tumors of this NAFLD-HCC model (new Supplemental Figure 5I). Whether ATG5 is causally involved in Nogo-B-induced NAFLD-associated hepatocarcinogenesis warrants further investigation by *in vivo* rescue experiments (lines 408-410).

Question 2. *Another major issue was the lack of experimental details in the figure legends in most figures or panels. It was unclear how many repeats were done and how many mice were used in some figures, which raised concerns on the rigor of the data and made it hard to evaluate.*

Response: We apologize for the obscurity. The numbers of mice have been added in the revised figure legends (Figures 2, 3 and 6). The *in vitro* experiments are presented as mean \pm SEM of 3 independent experiments as now indicated in both Methods and the figure legends.

Question 3. *The overall autophagy and lipophagy data were weak. Figure 5C need to be quantified. Only LC3-II levels were not sufficient to conclude autophagy activity. Figure 5D needs to add lysosomal inhibitors to assess the changes of LC3-II and p62. What was the molecular weight of ATG5 in figure 5? Was this the monomer of ATG5 or the conjugated form?*

Response: We thank the reviewer for the suggestions to strengthen our data. We have quantified the data in Figure 5C and depicted in new Supplemental Figure 5E. Upon treatment with the lysosomal inhibitor CQ, we found that Nogo-B-induced downregulation of p62 were abrogated in both LO2 and SMMC-7721 cells (new Figure 5D). The molecular weight of ATG5 in Figure 5 is 55-kD, indicating the ATG5-ATG12 conjugated form. Additionally, we found that monomeric ATG5 was also upregulated by Nogo-B (new Supplemental Figure 5F).

Question 4. *Figure 4, authors claimed increased LD staining after knockdown of Nogo-B in the starvation condition was due to the impaired lipophagy. However, it has been reported that knockdown of Atg genes or pharmacological inhibition of autophagy led to decreased LD numbers (Nguyen TB et al., Dev Cell 2017)? These controversial findings have not been discussed. Moreover, the area of lipid staining may not be the best way to quantify the lipid changes, the total cellular triglycerides should be used*

Response: We have discussed the roles of Nogo-B and ATG5 in LD degradation and formation in lines 366-374. Briefly, although mTORC1-regulated autophagy is reported to be necessary and sufficient for starvation-induced LD biogenesis, another study reported that the blockade of autophagy by pharmacological inhibition or silencing ATG5 caused a reduction in LD and TG breakdown^{3,4}. In our study, we

revealed the lipolytic role of Nogo-B in the liver is dependent on autophagy-mediated LD degradation instead of LD formation. Our data supports the role of autophagy in promoting lipid utilization and tumorigenesis.

We have also detected the levels of cellular triglycerides upon Nogo-B ectopic expression or knockdown. As shown in new Figure 4G, 4H, Supplemental Figure 4H and 4I, the amounts of cellular triglycerides were consistent with the lipid staining results.

Question 5. *Another major pathway to mediate LD lipolysis was the triglyceride lipase (ATGL) and hormone sensitive lipase (HSL). Does Nogo-B also affect ATGL and HSL?*

Response: We have measured the expressions of triglyceride lipase (*LIPE*) and hormone sensitive lipase (*PNPLA2*) in Nogo-B-overexpressing or knockdown cells. We found that Nogo-B did not affect their expressions (new Supplemental Figure 4L), suggesting that the role of Nogo-B may not involve LD lipolysis (lines 217-219).

Question 6. *Figure 4C, what was the lysate FFA and TG? In general, lipid should be extracted for the measurement of FFA and TG. This should be better clarified.*

Response: The “lysate TG” has been modified to “cellular TG”. We have measured the levels of FFA in the supernatants of Nogo-B-overexpressing and knockdown cells (new Figure 4G and 4H).

To Reviewer 3:

We appreciate the reviewer’s compliment on our “*striking and quite robust*” data. Now, we have performed additional experiments to strengthen the link between Nogo-B and lipophagy, such as better clarifying localization of LC3 and Nogo-B in LDs, and measuring secreted FFA. Additionally, we repeated some experiments to improve data quality and improved statistical analyses. The description has also amended according to reviewer’s suggestions, such as replacing ADFP use PLIN2, amending autophagosome to phagophore, and adding some detailed information on methodology.

Question 1. *Nogo-2 overexpression does not seem to increase autophagy flux as shown by the dual LC3 reporter. The number of red only dots (indicative of autolysosomes) seems to be similar to control in Figure S5B. Additionally, the*

authors show one cell for each group. Without quantification and statistics, the authors cannot make the claim that the differences are significant.

Response: We thank the reviewer for pointing out this issue. We have now quantified the number of dots and found a significant increase in both red and yellow dots in Nogo-B-expressing cells when compared to vector control cells (new Supplemental Figure 5D and 5E).

Question 2. *The images in Figure S5D are of poor quality. It would seem that there is 100% overlap between Nogo-B and ATG5, which seems difficult to believe and is more likely artifactual. Also, Nogo-B is an ER protein and the authors show it to be present on LD. The authors also show the association of autophagy proteins with Nogo-B. Now, is this interaction of Nogo-B occurring in the ER or at the LD? Is this an LD-ER contact site? Or is this simply ER-phagy and the lipids are inside the ER? The IP with the ER-motif mutant Nogo-B suggests the last option; however, additional experiments will be required to strengthen this connection.*

Response: We apologize for the obscurity. We have repeated immunostaining, which confirmed the co-localization of Nogo-B and ATG5 (new Figure 5G). The interaction between ER-bound Nogo-B and ATG5 is believed to occur in LDs that are formed at the ER membrane⁵. Once lipid droplets have formed, Nogo-B on the ER tubules will interact with the LDs' membrane. When LDs are degraded by lipophagy, the ATG5-containing phagophores engulf the LDs for clearance⁶. This issue has now been described in lines 356-358.

Question 3. *Does Nogo-B have LIR-motifs? It would be interesting to pursue this line if the authors want to make a stronger case of its link with autophagy.*

Response: We have checked the sequence of Nogo-B but found no LIR-motif.

Question 4. *An autophagosome does not have ATG5 on it. So the depiction in the proposed working model (Figure 7E) is incorrect.*

Response: The “autophagosome” has been modified to “phagophore” in the working model as shown in new Figure 7E.

Question 5. *Does Nogo-B induce the expression of autophagy genes – ATG5 and LC3?*

Response: We found that Nogo-B ectopic expression increased the expression of both conjugated ATG5-ATG12 and monomeric ATG5, as well as LC3 conversion in LO2 and SMMC-7721 cells (new Supplemental Figure 5F).

Question 6. *Instead of colocalizing Rab7 with LD, it would be more convincing to perform its colocalization with LC3 and LD in shNogo-B (Figure 5B), given the several non-autophagy roles of Rab7 and presence of Rab7 on other organelles.*

Response: Similar to RAB7 and LD, we have also detected the co-localization of LC3 and LD, which was significantly reduced in shNogo-B cells (new Figure 5B).

Question 7. *Figure 2B: Is Nogo-B present in the gene array?*

Response: Nogo-B was not present in the PCR Array for fatty liver-associated genes (Qiagen, PAHS-157Z) because it has not been previously linked to fatty liver.

Question 8. *Presence of Nogo-B on LD is shown in normal mouse livers. Is the protein present on the LD of HCC and NAFLD models as well?*

Response: We have performed Nogo-B immunostaining and Nile red staining in liver tissues from NAFLD and NAFLD-HCC. Although Nogo-B showed inverse correlation with LD due to promotion of LD degradation, we could still observe co-localization of Nogo-B and LD in some cells, especially in NAFLD livers (new Supplemental Figure 4C).

Question 9. *Nogo-B knockdown induced greater expression of LD marker ADFP (Figures S4B, S4C). However, no difference in the amount of LD was observed between shNogo-B and control cells in Figures 4E and 4F. Both legends say this was in response to oxLDL. This is an inconsistency.*

Response: We apologize for the mistake. The cells depicted in the original Supplemental Figure 4B and 4C should have been treated with oxLDL, followed by starvation to trigger LD degradation. The figure legends have been amended in new Supplemental Figure 4F and 4G.

Question 10. *Lysate FFAs are not a read-out of lipolysis. The cells generally don't have free fatty acids due to their toxicity.*

Response: Besides the cellular TG, we have further detected the FFA content in the supernatant, and found increased and decreased FFA levels by Nogo-B-overexpressing LO2 and SMMC-7721 cells and Nogo-B knockdown SK-Hep1 and MHCC97H cells, respectively (new Figure 4G, 4H and Supplemental Figure 4H and 4I).

Question 11. *The increase in TG upon KD of Nogo-B (Figure 4H) seems very modest. This does not match with the almost 3-fold increase in LD shown in Figure 4D even though the starvation time in Figure 4H is double that of Figure 4D (as per the legend).*

Response: We thought that the modest change in TG content may have been due to the sensitivity of the ELISA kit used in the original experiments. We thus measured the cellular TG using another ELISA kit, and found more robust differences in TG content between shCtrl and shNogo-B cells than originally reported (new Figure 4G, 4H and Supplemental Figure 4H and 4I). Additionally, lipid droplets contain a variety of lipids besides TG, and changes in accumulation of these lipids may also contribute to the enhanced LD content⁵.

Question 12. Figure 1K: the images do not show a decrease in volume/weight.

Response: We have added arrows to better indicate the tumors in shCtrl group, while the tumors in shNogo-B group are almost invisible in the image (new Figure 1K).

Question 13. Figure S4A: PLIN2 as LD marker is missing. ADFP should be replaced by PLIN2 as per current guidelines of naming perilipins/LD proteins.

Response: ADFP has been replaced by PLIN2, as suggested.

Question 14. Figure S4C: increased expression of PLIN2 (ADFP) in MHCC97H cells upon shNogo-B is not convincing.

Response: We have repeated the Western blot analysis and found increased PLIN2 expressions upon Nogo-B knockdown in both SK-Hep1 and MHCC97H cells (new Supplemental Figure 4G).

Question 15. Figure 4D: Resting shNogo-B control is missing.

Response: The result of oxLDL-loaded resting cells without starvation was similar in shNogo-B group as indicated in new Figure 4F, and now representative images are added in new supplemental Figure 4E.

Question 16. For all IF images, please include a nuclear stain to bring some perspective for the readers. It is difficult to identify the cell/s, for example, in 5A.

Response: We thank the reviewer for the suggestions to improve our data presentation. We have added the nuclear stain in all IF images as shown in new Figure 4B-4F, 4J, 5A, 5B, 5G and 5I.

Question 17. Some of the immunoblots are pixelated and some of the others are over-exposed.

Response: We have amended the immunoblots in new Figure 5E, 6G, 6I, 7A and Supplemental 4B and 4F.

Question 18. *The authors use ‘significantly’ without providing quantification and statistics.*

Response: To verify the significance of our findings, especially the immunofluorescence staining, we have quantified the signals and depicted the statistical difference as shown in new Figure 5B, 5G, Supplemental 2C, 4A, 4D, 5B, 5C, and 5E.

Question 19. *For LD isolation, the authors mention 3,300,000g centrifugation step with SW40 rotor. This is not possible. Please correct this error?*

Response: We apologize for the mistake. It should be 300,000g.

Question 20. *Most of the experiments with Nogo-B overexpression with Ctrl or sh ATG5 do not include the shATG5 only cells. In addition, the data showing that there is efficient KD of ATG5 with shRNA or siRNA is missing.*

Response: The effect of ATG5 knockdown on cell proliferation and LD numbers had been previously reported^{3,4,7} and now described in the discussion (lines 359-374). The efficient knockdown of ATG5 with shRNA is shown in new Supplemental Figure 5J.

Question 21. *The details regarding the plasmids used in this study are missing.*

Response: The plasmids information is now added in new supplemental Table 7.

Question 22. *The methodology for staining with LipidTOX is missing.*

Response: The methodology of LipidTOX staining has been added (lines 492-495).

Question 23. *The methodology for treatment of cells with ox-LDL is missing.*

Response: The methodology for ox-LDL treatment has been added (lines 490-492).

Question 24. *How ox-LDL is measured in serum is missing.*

Response: The ELISA kits used for measurement of oxLDL and LDL have been described in Methods (lines 528-529).

Question 25. *There are several typographical errors in the manuscript that must to be corrected.*

Response: The typographical errors in the manuscript have been corrected.

To Reviewer 4:

We appreciate the reviewer's compliment that "*the study is of interest*".

Question 1. *The initial array data shown in Suppl Fig 1A is not comprehensive as it only represents an average of 3 paired specimens. It would be useful to show each T and NT sample or each of the 3 ratio data to help determine the consistency of the finding. Is consistent data also seen in TCGA? The authors state so, but the data is not shown.*

Response: The T/NT ratio of each of the 3 samples have been shown in Figure 1B. The finding of the gene-of-interest Nogo-B is consistent with TCGA data as shown in Figure 1D.

Question 2. *In Fig 1A, how many HCC specimens in TCGA are interrogated? Are the HCC specimens used in TCGA for this analysis in Fig 1A or Fig 1D and Suppl Fig 1C all NAFLD-related or are they from different etiologies? The authors show later in the manuscript that Nogo-B is altered in HBV-HCC. If the latter, then it is difficult to relate Nogo-B effects specifically to lipid metabolism.*

Response: 360 HCC specimens in the TCGA are interrogated. The information of the HCC cohorts is now provided in Methods. Since only ~3-4% of the HCC patients have NAFLD, we have directly compared the *Nogo-B* expression between 16 and 20 pairs of NAFLD- and HBV-associated HCC tumor/non-tumor tissues, respectively. Quantitative RT-PCR analysis showed more significant upregulation of *Nogo-B* in NAFLD-associated HCC cases (Figure 2A), suggesting that *Nogo-B* may preferentially affect lipid metabolism.

Question 3. *The authors state that Nogo-B RNA copy number is upregulated in 3 cohorts but they do not show that copy number alterations are correlated or connected to expression changes of Nogo-B in the same samples. It is difficult to confirm its oncogenic function as the authors stet on line 106.*

Response: Re-analysis of TCGA database revealed that *Nogo-B* DNA copy number and mRNA expression exhibit a positive correlation (new Supplemental Figure 1D-1F). The two exome sequencing cohorts by Guichard *et al.* did not have accompanying RNA-seq data.

Question 4. *The authors screen several cell lines for Nogo-B level and identify two with lower levels and two with higher levels that are utilized for several experiments in the manuscript. However, in several cases, functional assays are only performed in one-representative cell line (e.g. Supp Fig 4E, 4G, 4H Fig 5B, H and I, 6C and D) and in Fig 5E anf F, the BEL-7404 cell line is used without rationale for not using either of the Nogo-B cell lines used in the previous figures.*

Response: We have performed new experiments using additional cell lines to validate our results in multiple cell lines, as shown in new Figure 5B, Supplemental 4H, 4I, 4K, 5K, 5L, 6B and 6D. BEL-7404 cell line was used as a representative line with relative high Nogo-B expression (Figure 1E). The endogenous interaction between Nogo-B and ATG5 has been further verified in MHCC97H cells (new Figure 5F).

Question 5. In Fig 6B, data from only 2 specimens in either the control of Nogo-B group are shown. A larger panel of specimens should be assayed to determine differences in relative lipidomic levels.

Response: Two specimens per group were used in the screening phase of lipidomics analysis. Sixteen specimens were later used to verify the findings as shown in Figure 6K and Supplemental Figure 6H.

Question 6. In Fig 7, data for only 8 and 4 pairs of NAFLD-HCC or HBV-HCC are shown, however 16 and 12 are noted in the manuscript text.

Response: Representative immunoblots of NAFLD-HCCs (8) and HBV-HCCs (4) were shown in Figure 7 from the total 16 and 12. We have now shown the data of the remaining 8 and 8 pairs of NAFLD-HCC and HBV-HCC in new Supplemental Figure 7.

Question 7. The association of the 3 genes with reduced patient survival has not been tested for confounding variables.

Response: The Cox analysis has been performed as shown in new Supplemental Table 8.

References

1. Dumas JF, Brisson L, Chevalier S, et al. Metabolic reprogramming in cancer cells, consequences on pH and tumour progression: Integrated therapeutic perspectives with dietary lipids as adjuvant to anticancer treatment. *Semin Cancer Biol* 2017;43:90-110.
2. Tian Y, Wong VW, Wong GL, et al. Histone Deacetylase HDAC8 Promotes Insulin Resistance and beta-Catenin Activation in NAFLD-Associated Hepatocellular Carcinoma. *Cancer Res* 2015;75:4803-1
3. Singh R, Kaushik S, Wang Y, et al. Autophagy regulates lipid metabolism. *Nature* 2009;458:1131-5.

4. Nguyen TB, Louie SM, Daniele JR, et al. DGAT1-Dependent Lipid Droplet Biogenesis Protects Mitochondrial Function during Starvation-Induced Autophagy. *Dev Cell* 2017;42:9-21 e5.
5. Mishra S, Khaddaj R, Cottier S, et al. Mature lipid droplets are accessible to ER luminal proteins. *J Cell Sci* 2016;129:3803-3815.
6. Martinez-Lopez N, Singh R. Autophagy and Lipid Droplets in the Liver, *Annu. Rev. Nutr.* 2015. 35:215–37.
7. Tian Y, Kuo CF, Sir D, et al. Autophagy inhibits oxidative stress and tumor suppressors to exert its dual effect on hepatocarcinogenesis. *Cell Death Differ* 2015;22:1025-34.

Reviewers' Comments:

Reviewer #1:

Remarks to the Author:

The authors addressed most of my questions properly and supported the answers by additional supplementary work.

Few minor comments remained:

Question 1: the authors added the numbers of HCC patients and controls. I suggest adding also more details about the patients' characteristics in a supplementary table. Also, please add the number of patients under figure 2B.

Question 5: The argument that additional breakdown of LD is necessary to support tumor growth (despite of the high availability of FA in NAFLD) due to the fact that mitochondrial fatty acid oxidation produces more ATP than oxidation of glucose is not taking into account the Warburg effect, which actually describes the opposite. One of the most known metabolic phenomenons of cancers cells is the preference to utilize glycolysis followed by cytosolic lactic acid fermentation despite the presence of adequate oxygen levels. This part of the discussion should therefore be replaced by other hypotheses.

Reviewer #2:

Remarks to the Author:

This is a revised manuscript to investigate an ER protein Nogo-B in NAFLD-associated liver tumorigenesis. Based on a large amount of in vitro and in vivo data the authors tended to conclude that increased oxLDL increased Nogo B that triggered lipophagy and YAP expression resulting in promoting liver tumorigenesis. While authors have performed some new experiments and the quality of the manuscript seemed to be improved, the data interpretation and conclusion on Nogo-B enhanced lipophagy that subsequently favored liver tumorigenesis remained somewhat weak and obscure. More convincing and solid data are needed given this is such an important impact of this paper if this is true.

Major specific comments:

1. As mentioned earlier in the first round review, the role of autophagy in lipophagy and the role of autophagy in liver tumorigenesis is still controversial, mice with liver-specific autophagy-defects (FIP200, Atg7 and Atg5) did not develop steatosis in response to fasting/refed or high fat diet. Therefore, at this stage, more convincing and solid data on how NogoB promotes autophagy and lipophagy are needed. In the revised manuscript, the lipid staining data were contradictory in Figure 3 and Figure 4. In figure 3, the mouse livers after high fat diet had decreased lipid staining in Nogo-B knockdown mice, but in figure 4, increased Nogo-B expression decreased lipid staining (Figure 4A, C). If Nogo-B indeed promoted autophagy and lipophagy, why would lipid decreased in Nogo-B knockdown mouse livers as the authors claimed that Nogo-B mediated autophagy to degrade LD? Better and more quantitative data on lipid contents should be provided in addition to the lipid staining (which is less quantitative and subjective). More autophagy data should also be provided in this in vivo mouse model.
2. The autophagy/lipophagy data in vitro remained weak. Figure 4B, the imaging data were saturated for GFP-NogoB, the pattern more reflect protein aggregates. Better image should be provided.
3. Figure 5A, the localization of LC3 with LD was impressive. However, it was unclear it was the LC3 protein itself recruited to LDs or it reflects autophagosomes enveloping LDs? It has been previously reported that LC3 protein can be recruited to LD to promote LD formation Data from electron microscopy studies of these cells would be very helpful.
4. Data from Figure 5C did not support increased autophagic flux by Nogo-B, but only suggested increased accumulation of autophagosomes. It was also unclear in this particular cell that presented here Nogo-B was overexpressed. There were also no quantification for these data. Figure 5D was a missing an important control group; the vector control cells treated with CQ, the

data were difficult to interpret without this important control.

5. Another new piece of exciting data was that Atg5 could interact with Nogo-B, and the Atg5 showed a nice ER pattern which has not been previously reported (Figure 5G). This striking finding should be validated using cellular fractionation to show the ER localization of Atg5 and Nogo-B together.

6. Another minor issue, please label the Atg5 as "Atg5-Atg12" if the band detected was at 55KD, otherwise it is confusing.

Reviewer #3:

Remarks to the Author:

In the revised version the authors have addressed most of my original concerns.

Fig S5D and S5E. The images are still poor quality and show one cell each. Additionally, only SMMC-7721 cells are shown and not LO2 cell, whereas the graphs are shown for both cell types. There are many more panels in other figures with one or two cells only. At times it is difficult to appreciate the differences the authors are claiming.

The vector is mCherry-EGFP-LC3. The authors refer to it as LC3-mCherry-GFP or variants of it. The authors should please correct this. The authors also refer to GFP+ at times. Please refer to the plasmid correctly and consistently.

ATG5-ATG12 is approximately 55kda while the ATG5 monomer is approximately 33kda. The monomeric ATG5 band the authors depict in Figure S5F is too close to the conjugate. Is this really the monomer or a non-specific band? Absence of MW ladder makes it difficult to say. Additionally, the ATG5 antibody recognizes the ATG5-ATG12 conjugate. The authors should label that band as ATG5-ATG12 and not ATG5.

New Figure 5B. What does colocalization area per cell mean for Rab7 alone, LD alone, LC3 alone?

The authors refer often to "semi-quantitation". What does that refer to?

The IF images are often saturated, and hence, not accurately quantifiable. Especially, Figure 5A and Figure 5B (which has been quantified).

Figure S4A: PLIN2 as LD marker is still missing.

For LD isolation, the authors mentioned in the original manuscript 3,300,000g centrifugation step with SW40 rotor. This is not possible. Now, in the revision, they mention 300,000g. That is still impossible. What did the authors use for the LD isolation? Additionally, in the methods, the authors state, "The white band containing lipid droplets at the top of gradient was collected in 0.5 mL, then the sample was centrifuged at 20,000 g for 3 min, the underlying solution was carefully removed, and droplets were gently resuspended in 200 μ L of buffer B for Western blot analysis." My question is: what are the authors taking for their Western blot analysis of droplets – the underlying solution?

The reviewer asked in the original manuscript for the details for the LIPIDTOX staining. The authors have not provided the details. The concentration used for the staining of lipid droplets should be mentioned in the methods.

Reviewer #4:

Remarks to the Author:

The authors have made a concerted effort to respond to prior review of the original manuscript. However, in the revised version, the data provided in Suppl Fig 7 do not seem to fully support the statement in the manuscript that "Compared with the paired adjacent normal liver tissues, concordant upregulation of Nogo-B, oxLDL, CD36, and CEBP β and downregulation of p-YAP were 308 detected in NAFLD-associated HCC tissues". In specimens 11, 14 and 16 for example, the level of Nogo-B is lower in T vs NT and is not fully consistent with other proteins (e.g. CD36). In addition, the figure seems to be a compilation of blots which makes it difficult to ascertain the consistency of the data.

In addition, multivariable analysis shown in Suppl Table 8 should be performed with the referent group as the group with least risk and should be categorized this way across all variables. In the case of Nogo-B this should be the "low" group and for gender, this should be the female group (high vs low; Male vs female). Stage and cirrhosis for example are categorized inversely. This analysis should be re-done. There should also be a rationale for why certain cutoffs are used, for example an AFP cutoff of 500 and units should be provided. There should also be a notation for whether a test of the proportionality assumption was made and whether the data passed that test. The column header states that a hazard ratio and confidence interval are shown however, only the hazard ratio is actually shown. Confidence interval for each variable should be shown.

Point-to-point responses to Reviewers

Reviewer 1

Ref 1.1. Characteristics of HCC patients

Reviewer Comment	The authors addressed most of my questions properly and supported the answers by additional supplementary work. Few minor comments remained: Question 1: the authors added the numbers of HCC patients and controls. I suggest adding also more details about the patients' characteristics in a supplementary table. Also, please add the number of patients under figure 2B.
Author Response	We have added the characteristics of NAFLD-associated HCC patients in the revised Supplemental Table 8, and the number of samples in the figure legend of Figure 2B.

Ref 1.2. Hypothesis on the metabolic phenomenon

Reviewer Comment	Question 5: The argument that additional breakdown of LD is necessary to support tumor growth (despite of the high availability of FA in NAFLD) due to the fact that mitochondrial fatty acid oxidation produces more ATP than oxidation of glucose is not taking into account the Warburg effect, which actually describes the opposite. One of the most known metabolic phenomena of cancer cells is the preference to utilize glycolysis followed by cytosolic lactic acid fermentation despite the presence of adequate oxygen levels. This part of the discussion should therefore be replaced by other hypotheses.
Author Response	We thank the reviewer's important comment on the metabolic phenomenon of cancer cells. Cancer cells are exposed to intermittent hypoxic episodes in the acidic microenvironment which interferes with the metabolism of oxygenated cancer cells. Two recent studies have demonstrated that cancer cells subjected to an acidic environment have a preferred glutamine reductive metabolism (Ref 1), and fatty acid oxidation to provide acetyl-coA to the tricarboxylic cycle. The consequences of this rewiring of metabolism are the preferred consumption of fatty acids to provide energy, and also the reduction of ROS production, which together support cancer cell proliferation and tumor growth (Ref 2). In the absence of available nutrients, cancer cells can withstand long periods of nutrient deprivation via the self-catabolic process of macroautophagy to liberate free amino and fatty acids (Ref 3). Lipophagy, as one of the main types of macroautophagy in cancer cells, could supply fatty acids for cell survival. This new discussion is added on P.15 of the revised manuscript.

Reviewer 2

Ref 2.1. Additional data on NogoB and autophagy

Reviewer Comment	This is a revised manuscript to investigate an ER protein Nogo-B in NAFLD-associated liver tumorigenesis. Based on a large amount of in vitro and in vivo data the authors tended to conclude that increased oxLDL increased NogoB that triggered lipophagy and YAP expression resulting in promoting liver tumorigenesis. While authors have performed some new experiments and the quality of the manuscript seemed to be improved, the data interpretation and conclusion on Nogo-B enhanced lipophagy that subsequently favored liver tumorigenesis remained somewhat weak and obscure. More convincing and solid data are needed given this is such an important impact of this paper if this is true. Major specific comments: 1. As mentioned earlier in the first round review, the role of autophagy in lipophagy and the role of autophagy in liver tumorigenesis is still controversial, mice with liver-specific autophagy-defects (FIP200, Atg7 and Atg5) did not develop steatosis in response to fasting/refed or high fat diet. Therefore, at this stage, more convincing and solid data on how NogoB promotes autophagy and lipophagy are needed. In the revised manuscript, the lipid staining data were contradictory in Figure 3 and Figure 4. In figure 3, the mouse livers after high fat diet had decreased lipid staining in Nogo-B knockdown mice, but in figure 4, increased Nogo-B expression decreased lipid staining (Figure 4A, C). If Nogo-B indeed promoted autophagy and lipophagy, why would lipid decreased in Nogo-B knockdown mouse livers as the authors claimed that Nogo-B mediated autophagy to degrade LD? Better and more quantitative data on lipid contents should be provided in addition to the lipid staining (which is less quantitative and subjective). More autophagy data should also be provided in this in vivo mouse model.
Author Response	We thank the reviewer's important comment on the role of Nogo-B on autophagy in HCC, and concur with the view on the paradoxical role of autophagy in cancers including HCC (Ref 4 and 5). As suggested by the reviewer, we have performed additional quantitative analysis of lipid contents in the NAFLD-associated HCC mouse model. As the ELISA results shown in the new Supplemental Figure 3C, the hepatic TG, FFA, cholesterol, LPA and oxLDL contents were concomitantly increased in the HFHC dietary NAFLD-associated HCC model. Notably, knockdown of Nogo-B in the livers of obese mice significantly reduced the levels of these lipids except oxLDL. We have also provided new autophagy data from Western blot analysis of liver tissues in the NAFLD-associated HCC mouse model. In concordance with elevated Nogo-B expression, we found higher Atg5-Atg12 and LC3-II but lower p62 levels in the HFHC diet as compared to the control diet group (new Supplemental Figure 5M), and the changes of autophagy markers were remarkably reversed by Nogo-B downregulation. To further clarify the autophagy level in NAFLD-associated HCC, we performed Western blot using the tumorous and adjacent non-tumorous tissues of the mouse model. Consistently, HCC tumors exhibited higher Atg5-Atg12 and LC3-II, and lower p62 levels than non-tumors (new Supplemental Figure 5N). These results consolidate our observation that Nogo-B promotes autophagy in NAFLD-HCC development. The decreased lipid contents in Nogo-B knockdown mouse livers are likely systemic effects as a result of crosstalk between autophagy and metabolic/oncogenic signaling pathways (Ref 4-6). In addition, Nogo-B-activated YAP signaling has been reported to

	promote hepatic steatosis and HCC (Ref 7 and 8). Therefore, it is conceivable to speculate that YAP inactivation by Nogo-B knockdown might alleviate lipid accumulation in NAFLD progression. The Nogo-B-YAP pathway could be further subverted by cancer cells for metabolic reprogramming in the later stages of hepatocarcinogenesis. These new results are added on P.7 and P.10 of the revised manuscript.
--	---

Ref 2.2. Improvement of image quality

Reviewer Comment	2. The autophagy/lipophagy data in vitro remained weak. Figure 4B, the imaging data were saturated for GFP-NogoB, the pattern more reflect protein aggregates. Better image should be provided.
Author Response	We have now provided clearer images which show co-localization of punctate Nogo-B and lipids (new Figure 4B).

Ref 2.3. Localization of LC3 with LD

Reviewer Comment	3. Figure 5A, the localization of LC3 with LD was impressive. However, it was unclear it was the LC3 protein itself recruited to LDs or it reflects autophagosomes enveloping LDs? It has been previously reported that LC3 protein can be recruited to LD to promote LD formation Data from electron microscopy studies of these cells would be very helpful.
Author Response	We have performed electron microscopy as suggested and found double-membrane vesicles analogous to autophagosomes (arrow heads) around LDs (arrows) and degradative structures enriched in LDs (asterisks, new Supplemental Figure 5C). This result suggests a notion of autophagosomes enveloping LDs as described in a previous report (Ref 9).

Ref 2.4. Autophagic flux by Nogo-B

Reviewer Comment	4. Data from Figure 5C did not support increased autophagic flux by Nogo-B, but only suggested increased accumulation of autophagosomes. It was also unclear in this particular cell that presented here Nogo-B was overexpressed. There were also no quantification for these data. Figure 5D was a missing an important control group; the vector control cells treated with CQ, the data were difficult to interpret without this important control.
Author Response	We have repeated this experiment using Nogo-B-stably-over-expressing and vector control cell lines. As shown in new Figure 5C and new Supplemental Figure 5E-G, the numbers of both autophagosomes (mCherry⁺, EGFP⁺) and autolysosomes (mCherry⁺, EGFP⁻) were significantly increased by Nogo-B over-expression, as quantified in new Supplemental Figure 5F. We have also included new control groups i.e. vector control cells treated with CQ and

	wortmannin, an early-stage autophagy inhibitor (new Figure 5D and new Supplemental Figure 5H). The data support the notion that Nogo-B promotes autophagic flux in HCC cells.
--	---

Ref 2.5. ER localization of Atg5 and Nogo-B

Reviewer Comment	5. Another new piece of exciting data was that Atg5 could interact with Nogo-B, and the Atg5 showed a nice ER pattern which has not been previously reported (Figure 5G). This striking finding should be validated using cellular fractionation to show the ER localization of Atg5 and Nogo-B together.
Author Response	We thank the reviewer for the compliment. To clarify the location of Atg5 and Nogo-B interaction, we performed cellular fractionation as suggested and found Atg5-Atg12 and Nogo-B co-expression in ER and cytoplasm, but not nucleus (new Supplemental Figure 5L), which is consistent with the immunofluorescence data (Figure 5G).

Ref 2.6. Proper labeling

Reviewer Comment	6. Another minor issue, please label the Atg5 as "Atg5-Atg12" if the band detected was at 55KD, otherwise it is confusing.
Author Response	We have amended the labels accordingly.

Reviewer 3

Ref 3.1. Improvement of image quality

Reviewer Comment	In the revised version the authors have addressed most of my original concerns. Fig S5D and S5E. The images are still poor quality and show one cell each. Additionally, only SMMC-7721 cells are shown and not LO2 cell, whereas the graphs are shown for both cell types. There are many more panels in other figures with one or two cells only. At times it is difficult to appreciate the differences the authors are claiming.
Author Response	We have provided new images showing more cells (LO2 in new Figure 5C and SMMC-7721 in new Supplemental Figure 5E), and quantified the data as shown in Supplemental Figure 5F. Other images e.g. Figure 5B have also been replaced to better demonstrate the differences.

Ref 3.2. Correction of plasmid names

Reviewer Comment	The vector is mCherry-EGFP-LC3. The authors refer to it as LC3-mCherry-GFP or variants of it. The authors should please correct this. The authors also refer to GFP+ at times. Please refer to the plasmid correctly and consistently.
------------------	---

Author Response	We are sorry for the confusion. The plasmid names have been corrected accordingly.
-----------------	--

Ref 3.3. Labeling of molecular weight

Reviewer Comment	ATG5-ATG12 is approximately 55kda while the ATG5 monomer is approximately 33kda. The monomeric ATG5 band the authors depict in Figure S5F is too close to the conjugate. Is this really the monomer or a non-specific band? Absence of MW ladder makes it difficult to say. Additionally, the ATG5 antibody recognizes the ATG5-ATG12 conjugate. The authors should label that band as ATG5-ATG12 and not ATG5.
Author Response	We have included the molecular weights (new Supplemental Figure 5I) to indicate the monomeric and conjugated forms of ATG5.

Ref 3.4. Labeling of axis

Reviewer Comment	New Figure 5B. What does colocalization area per cell mean for Rab7 alone, LD alone, LC3 alone?
Author Response	We have amended the axis as 'Positive staining area per cell'.

Ref 3.5. Meaning of term

Reviewer Comment	The authors refer often to "semi-quantitation". What does that refer to?
Author Response	It was actually meant 'quantification'. We are sorry for the confusion.

Ref 3.6. Improvement of image quality

Reviewer Comment	The IF images are often saturated, and hence, not accurately quantifiable. Especially, Figure 5A and Figure 5B (which has been quantified).
Author Response	We thank the reviewer's suggestion to improve the image quality, and have reduced the signal levels to avoid saturation of the IF images.

Ref 3.7. Missing of marker

Reviewer Comment	Figure S4A: PLIN2 as LD marker is still missing.
Author	We are sorry for missing the marker, which has now been added in the new

Response	Supplemental Figure 4B.
----------	-------------------------

Ref 3.8. Clarification of method

Reviewer Comment	For LD isolation, the authors mentioned in the original manuscript 3,300,000g centrifugation step with SW40 rotor. This is not possible. Now, in the revision, they mention 300,000g. That is still impossible. What did the authors use for the LD isolation? Additionally, in the methods, the authors state, "The white band containing lipid droplets at the top of gradient was collected in 0.5 mL, then the sample was centrifuged at 20,000 g for 3 min, the underlying solution was carefully removed, and droplets were gently resuspended in 200 µL of buffer B for Western blot analysis." My question is: what are the authors taking for their Western blot analysis of droplets – the underlying solution?
Author Response	We are very sorry for this mistake again. We used 38000 rpm for LD isolation as mentioned previously (Ref 10). Since the liquid in the centrifuge tube was full when we isolated LD. So we should use average centrifugal radius (r_{av}) for the conversion from rpm to rcf. The r_{av} of SW40 Ti rotor is 112.7 mm and the results of calculation using formula '$rcf = 1.12r (rpm/1000)^2$' is about 180,000g. We are also very sorry for our unclear and confusing description of the LD isolation method. As previous report described (Ref 10), the sample was centrifuged at 20,000 g to separate the LDs from the buffer. Then the underlying solution and pellet was removed and discarded using a gel-loading tip. After that, the upper layer which contained LDs was resuspended in 200 µL of buffer B, and further washing and centrifugation steps were repeated twice as described above. The upper layer (white band) of last centrifugation which is much purer LDs was used for Western blot analysis of lipid droplets. This description has now been amended in methods.

Ref 3.9. Clarification of method

Reviewer Comment	The reviewer asked in the original manuscript for the details for the LIPIDTOX staining. The authors have not provided the details. The concentration used for the staining of lipid droplets should be mentioned in the methods.
Author Response	The LIPIDTOX concentration for staining is 1:1000 as recommended by the manufacturer, and has now been added in the Methods.

Reviewer 4

Ref 4.1. Consistency of data from clinical specimens

Reviewer Comment	The authors have made a concerted effort to respond to prior review of the original manuscript. However, in the revised version, the data provided in Suppl Fig 7 do not seem to fully support the statement in the manuscript that "Compared with the paired adjacent normal liver tissues, concordant upregulation of Nogo-B, oxLDL, CD36, and
------------------	---

	CEBPβ and downregulation of p-YAP were 308 detected in NAFLD-associated HCC tissues". In specimens 11, 14 and 16 for example, the level of Nogo-B is lower in T vs NT and is not fully consistent with other proteins (e.g. CD36). In addition, the figure seems to be a compilation of blots which makes it difficult to ascertain the consistency of the data.
Author Response	We agree with the reviewer that there is inconsistency of data in a few cases of the clinical specimens, which is typical given the various backgrounds of the patients. Thus we have now provided the characteristics of NAFLD-associated HCC patients in the revised Supplemental Table 8. Although there is variation among individual samples, there is overall statistical significance in both the expressions between groups (Figure 7B) and the correlations between related molecules (Figure 7C). We have amended the data description as "Compared with the paired adjacent normal liver tissues, concordant upregulation of Nogo-B, oxLDL, CD36, and CEBPβ and downregulation of p-YAP were detected in most NAFLD-associated HCC tissues" .

Ref 4.2. Statistical analysis

Reviewer Comment	In addition, multivariable analysis shown in Suppl Table 8 should be performed with the referent group as the group with least risk and should be categorized this way across all variables. In the case of Nogo-B this should be the "low" group and for gender, this should be the female group (high vs low; Male vs female). Stage and cirrhosis for example are categorized inversely. This analysis should be re-done. There should also be a rationale for why certain cutoffs are used, for example an AFP cutoff of 500 and units should be provided. There should also be a notation for whether a test of the proportionality assumption was made and whether the data passed that test. The column header states that a hazard ratio and confidence interval are shown however, only the hazard ratio is actually shown. Confidence interval for each variable should be shown.
Author Response	We thank the reviewer's suggestion on statistical analysis. We have re-done the Cox analysis using the lowest risk group as the reference group and added the confidence interval for each variable (Supplemental Table 9). The cutoff of each variable was chosen as mentioned previously (Ref 11). We have also used the R function <code>cox.zph</code> to test the proportionality assumption. In both univariate (Supplemental Table 10) and multivariate analyses (Supplemental Table 11), all the variables have passed the test.

References

1. Corbet C, Draoui N, Polet F, Pinto A, Drozak X, Riant O, Feron O. The SIRT1/HIF2 alpha axis drives reductive glutamine metabolism under chronic acidosis and alters tumor response to therapy. *Cancer Res.* 2014; 74:5507–5519.
2. Corbet C, Pinto A, Martherus R, Santiago de Jesus JP, Polet F, Feron O. Acidosis drives the reprogramming of fatty acid metabolism in cancer cells through changes in mitochondrial and histone acetylation. *Cell Metab.* 2016; 24:311–323.

3. Boya P, Reggiori F, Codogno P. Emerging regulation and functions of autophagy. *Nat. Cell Biol.* 2013; 15:713-720.
4. Wu WK, Coffelt SB, Cho CH, Wang XJ, Lee CW, Chan FK, et al. The autophagic paradox in cancer therapy. *Oncogene.* 2012; 31:939-53.
5. Wu WK, Zhang L, Chan MTV. Autophagy, NAFLD and NAFLD-related HCC. *Adv Exp Med Biol.* 2018; 1061:127-138.
6. Singh R, Kaushik S, Wang Y, Xiang Y, Novak I, Komatsu M, et al. Autophagy regulates lipid metabolism. *Nature.* 2009; 458:1131-5.
7. Jeong SH, Kim HB, Kim MC, Lee JM, Lee JH, Kim JH, Kim JW, Park WY, Kim SY, Kim JB, Kim H, Kim JM, Choi HS, Lim DS. Hippo-mediated suppression of IRS2/AKT signaling prevents hepatic steatosis and liver cancer. *J Clin Invest.* 2018; 128:1010-1025.
8. Chen P, Luo Q, Huang C, Gao Q, Li L, Chen J, Chen B, Liu W, Zeng W, Chen Z. Pathogenesis of non-alcoholic fatty liver disease mediated by YAP. *Hepatol Int.* 2018; 12:26-36.
9. Ouimet M, Franklin V, Mak E, Liao X, Tabas I, Marcel YL. Autophagy regulates cholesterol efflux from macrophage foam cells via lysosomal acid lipase. *Cell Metab.* 2011; 13:655-67.
10. Ding Y, Zhang S, Yang L, Na H, Zhang P, Zhang H, Wang Y, Chen Y, Yu J, Huo C, Xu S, Garaiova M, Cong Y, Liu P. Isolating lipid droplets from multiple species. *Nat Protoc.* 2013; 8:43-51.
11. Yang Y, Zhou Y, Hou J, Bai C, Li Z, Fan J, Ng IOL, Zhou W, Sun H, Dong Q, Lee JMF, Lo CM, Man K, Yang Y, Li N, Ding G, Yu Y, Cao X. Hepatic IFIT3 Predicts Interferon-a Therapeutic Response in Patients of Hepatocellular Carcinoma. *Hepatology.* 2017; 66:152-166.

Reviewers' Comments:

Reviewer #1:

Remarks to the Author:

The authors addressed all my comments properly.

Reviewer #2:

Remarks to the Author:

Specific comments:

1. The concerns on the data rigor and robustness remained. Many figures remained only showed one cell for the imaging studies, for instance, fig 5A and C; Sfig 2, sfig 5D and E;
2. Authors are strongly recommended to add all the molecular weight to all the blots. What was the size of ATG5-ATG12? This antibody needs to be validated using Atg5 Knockout cells (Atg5 KO MEF are easily obtained). Figure 5G, Atg5 staining pattern looks like cytoskeleton structure, which somewhat supporting the concern on the specificity of Atg5 antibody used in this study.
3. Sfig5M, decreased autophagy proteins and autophagy activity has been reported in many studies, thus the increased Atg5-Atg12 again supported the concern on this antibody? Did the authors observe increase of other ATG genes?
4. Figure 3D, Nile red staining of liver tissues in shRNA NogoB treated mice appeared to be less than shcontrol mice, which is contradictory to the conclusion that NogoB induced lipophagy.

Reviewer #3:

Remarks to the Author:

The authors have addressed my comments in this round of review.

Reviewer #4:

Remarks to the Author:

While the authors have modified Suppl Table 8, information regarding assumption testing should be added to the statistical methods section of the manuscript.

While the authors have provided some additional clarification regarding the status of protein levels, the data shown in Suppl Fig 7 remain a compilation of blots as noted in the prior review, making it difficult to ascertain the consistency of the data.

Point-to-point responses

Reviewer #2:

1. *The concerns on the data rigor and robustness remained. Many figures remained only showed one cell for the imaging studies, for instance, fig 5A and C; Sfig 2, sfig 5D and E;*

Response: We have now provided images of multiple cells in the new Figure 5A, supplemental Figure 2C and supplemental Figure 5D. The original images in Figure 5C and supplemental Figure 5E had included more than 3 cells.

2. *Authors are strongly recommended to add all the molecular weight to all the blots. What was the size of ATG5-ATG12? This antibody needs to be validated using Atg5 Knockout cells (Atg5 KO MEF are easily obtained). Figure 5G, Atg5 staining pattern looks like cytoskeleton structure, which somewhat supporting the concern on the specificity of Atg5 antibody used in this study.*

Response: We thank the reviewer for this constructive suggestion to increase data rigor. We have added the molecular weight to all blots. The size of ATG5-ATG12 is 55KD (supplemental Figure 5I). As suggested, the antibody has been validated using Atg5 KO MEF cells as shown below. The MEF cell line was kindly provided by Prof. Hong Zhang and Dr. Guangyan Miao (IBP, CAS). Atg5 KO MEFs were established by immortalizing Atg5^{fl/fl} MEF from Atg5^{fl/fl} mice by transfecting with pEF321-T containing the SV40 large T antigen. The cells were then transiently transfected with pCre-Pac and Atg5 KO MEFs were selected with 2.5 µg/ml puromycin. Single knockout clones were verified by immunoblotting with anti-Atg5 antibody. Both WT and Atg5 KO MEFs are female.

3. *Sfig5M, decreased autophagy proteins and autophagy activity has been reported in many studies, thus the increased Atg5-Atg12 again supported the concern on this antibody? Did the authors observe increase of other ATG genes?*

Response: In addition to Atg5-Atg12, we have observed increased Atg7 expression in the HFHC compared to the control group (new supplemental Figure 5M). Moreover, we also detected

increased expression of ATG7 in Nogo-B-expressing cell lines and tumor samples (new supplemental Figure 5I and 5N).

4. *Figure 3D, Nile red staining of liver tissues in shRNA NogoB treated mice appeared to be less than shcontrol mice, which is contradictory to the conclusion that NogoB induced lipophagy.*

Response: As addressed in our previous Response Letter, the decreased lipid contents in Nogo-B knockdown mouse livers are likely systemic effects as a result of crosstalk between autophagy and metabolic/oncogenic signaling pathways (Ref 1-3). In addition, consistent with our results on Nogo-B-induced YAP activation, elevated YAP signaling has been reported to promote hepatic steatosis and HCC (Ref 4-5). Therefore, it is conceivable to speculate that YAP inactivation by Nogo-B knockdown (Figure 6H, 6I, supplemental Figure 6F and new supplemental Figure 6G) might alleviate lipid accumulation as observed in Figure 3D.

Reviewer #4:

1. *While the authors have modified Suppl Table 8, information regarding assumption testing should be added to the statistical methods section of the manuscript.*

Response: The information regarding assumption testing has been added to the statistical methods section of the manuscript.

2. *While the authors have provided some additional clarification regarding the status of protein levels, the data shown in Suppl Fig 7 remain a compilation of blots as noted in the prior review, making it difficult to ascertain the consistency of the data.*

Response: We thank the reviewer's important comment on the clinical data quality, especially in supplemental Figure 7. We repeated the Western blot analysis of unclear markers Nogo-B, CEBP β , p-YAP and YAP in last 6 pairs of tissue, and now provided the uncut full western blots gels for all clinical specimens in new supplemental Figure 8. We have also included the quantification data of the bands of Nogo-B, oxLDL, CD36, CEBP β , and p-YAP in new Figure 7A and supplemental Figure 7.

References:

1. Singh R, Kaushik S, Wang Y, Xiang Y, Novak I, Komatsu M, et al. Autophagy regulates lipid metabolism. *Nature*. 2009; 458:1131–5.
2. Wu WK, Coffelt SB, Cho CH, Wang XJ, Lee CW, Chan FK, et al. The autophagic paradox in cancer therapy. *Oncogene*. 2012; 31:939–53.

3. Wu WK, Zhang L, Chan MTV. Autophagy, NAFLD and NAFLD-related HCC. *Adv Exp Med Biol.* 2018; 1061:127-138.
4. Jeong SH, Kim HB, Kim MC, Lee JM, Lee JH, Kim JH, Kim JW, Park WY, Kim SY, Kim JB, Kim H, Kim JM, Choi HS, Lim DS. Hippo-mediated suppression of IRS2/AKT signaling prevents hepatic steatosis and liver cancer. *J Clin Invest.* 2018; 128:1010-1025.
5. Chen P, Luo Q, Huang C, Gao Q, Li L, Chen J, Chen B, Liu W, Zeng W, Chen Z. Pathogenesis of non-alcoholic fatty liver disease mediated by YAP. *Hepatol Int.* 2018; 12:26-36.

Reviewers' Comments:

Reviewer #2:

Remarks to the Author:

I am satisfied with the revision

Reviewer #4:

Remarks to the Author:

The authors have satisfactorily responded to the items in the previous round of review.